# Model-based analysis uncovers mutations altering autophagy selectivity in human cancer

Zhu Han[1,7], Weizhi Zhang[2,7], Wanshan Ning [2], Chenwei Wang[2], Wankun Deng[2], Zhidan Li [1], Zehua Shang [1], Xiaofei Shen[3], Xiaohui Liu[4], Otto Baba[5], Tsuyoshi Morita [5], Lu Chen[1], Yu Xue [2,6✉] & Da Jia [1✉]

Autophagy can selectively target protein aggregates, pathogens, and dysfunctional organelles for the lysosomal degradation. Aberrant regulation of autophagy promotes tumorigenesis, while it is far less clear whether and how tumor-specific alterations result in autophagic aberrance. To form a link between aberrant autophagy selectivity and human cancer, we establish a computational pipeline and prioritize 222 potential LIR (LC3-interacting region) motif-associated mutations (LAMs) in 148 proteins. We validate LAMs in multiple proteins including ATG4B, STBD1, EHMT2 and BRAF that impair their interactions with LC3 and autophagy activities. Using a combination of transcriptomic, metabolomic and additional experimental assays, we show that STBD1, a poorly-characterized protein, inhibits tumor growth via modulating glycogen autophagy, while a patient-derived W203C mutation on LIR abolishes its cancer inhibitory function. This work suggests that altered autophagy selectivity is a frequently-used mechanism by cancer cells to survive during various stresses, and provides a framework to discover additional autophagy-related pathways that influence carcinogenesis.

[1] Key Laboratory of Birth Defects and Related Diseases of Women and Children, Department of Paediatrics, West China Second University Hospital, State Key Laboratory of Biotherapy and Collaborative Innovation Center of Biotherapy, Sichuan University, Chengdu, China. [2] Key Laboratory of Molecular Biophysics of Ministry of Education, Hubei Bioinformatics and Molecular Imaging Key Laboratory, Center for Artificial Intelligence Biology, College of Life Science and Technology, Huazhong University of Science and Technology, Wuhan, Hubei, China. [3] Hospital of Chengdu University of Traditional Chinese Medicine, Chengdu, China. [4] School of Life Sciences, Tsinghua University, Beijing, China. [5] Oral and Maxillofacial Anatomy, Tokushima University Graduate School, Kuramoto-Cho, Tokushima, Japan. [6] Nanjing University Institute of Artificial Intelligence Biomedicine, Nanjing, Jiangsu, China. [7] These authors contributed equally: Zhu Han, Weizhi Zhang. ✉email: xueyu@hust.edu.cn; Jiada@scu.edu.cn

Macroautophagy (hereafter referred to as autophagy) is an evolutionarily conserved catabolic process and is characterized by the formation of double-membrane vesicles known as autophagosomes[1,2]. Whereas autophagy occurs at a basal level in all cells, it is induced by many extracellular and intracellular stimuli[3]. In addition to starvation-induced bulk autophagy, autophagy can also selectively target many parts of cells as cargoes for degradation, ranging from damaged organelles to pathogens inside vacuoles or the cytosol, from misfolded proteins to specific inflammatory signaling molecules[1,2]. Thus, autophagy plays diverse functions in cells and is critical for maintaining cellular, tissue, and organismal homeostasis. Dysregulation of autophagy has been linked to the pathogenesis of a broad range of diseases, in particular cancer, neurodegenerative diseases, and metabolic diseases[1,2].

Cargoes are targeted by selective autophagy through a variety of mechanisms, including utilizing the LC3-interacting region (LIR) motif, ubiquitin-interacting motif, or binding to the TRIM family proteins[4–8]. Among them, the LIR motif, also named as ATG8-interaction motif (AIM), is the best-characterized one. LIR is a short peptide sequence binding to members of the Atg8 family, comprising the microtubule-associated protein 1 light chain 3B (MAP1LC3B/LC3) analogs or γ-aminobutyric acid-receptor associated proteins (GABARAPs)[5,6]. In addition to substrates for selective autophagy, many autophagy-related (ATG) proteins and autophagy regulators also contain the LIR motif[5,6,9]. Thus, the LIR–LC3 interaction is essential for the formation, transport, and maturation of autophagosomes. Genetic mutations in LIR motifs may significantly alter the binding affinity to LC3, thereby altering the autophagy selectivity and contributing to the pathogenesis of multiple diseases, such as neurological disorders. For example, an L341V missense mutation found in the LIR motif of sequestosome 1 (SQSTM1/p62), identified in a patient with sporadic amyotrophic lateral sclerosis, disrupts the binding to LC3B and inhibits p62 recruitment into autophagosomes[10].

The involvement of autophagy in tumor pathogenesis is well-established, which may promote or suppress carcinogenesis depending on the cancer type and stage. Activation of autophagy enables cancer cells to survive under stresses, including nutrient deprivation, hypoxia, or anti-cancer treatment[11]. However, suppression of autophagy can also promote tumorigenesis through accumulating genotoxic cellular wastes and facilitating additional genomic mutations[12,13]. The dual functions of autophagy in cancer pathogenesis is also supported by the analysis of the genome, transcriptome, and proteome of human cancer samples, which revealed many recurrently altered ATG genes and autophagy regulators in human tumors[14,15]. Despite these efforts, it remains unknown whether DNA alterations present in the cancer patient samples lead to changes in autophagy selectivity, and how cancer cells benefit from these changes.

We hypothesize that a subset of human cancer mutations may alter autophagy selectivity by impacting the LIR motif. Thus, analysis of the mutations will not only confirm the roles of ATG genes and autophagy regulators in various cancers but also discover new autophagy pathways that contribute to carcinogenesis. To explore the link between aberrant autophagy selectivity and human cancer, we develop a pipeline named "inference of cancer-associated LIR-containing proteins" (iCAL), which integrates a new algorithm named "prediction of the LIR motif" (pLIRm), a model-based algorithm named pLAM to predict LIR motif-associated mutations (LAMs), a pan-cancer analysis, and cell- and animal-based validations. Using iCAL, we have identified 148 LIR-containing proteins (LIRCPs) that carry single point mutations within the LIR motif, including some well-established ATG genes and autophagy regulators as well as many novel candidate genes. Among these candidate genes, we functionally confirm that starch-binding domain-containing protein 1 (STBD1), a gene involved in transporting glycogen to lysosomes, has a previously unappreciated role in suppressing cancer growth. Mechanistically, STBD1 inhibits tumor growth via metabolic reprogramming in cancer cells, including rewiring glycolysis and the pentose phosphate pathway. Thus, our study provides an integrative approach to discover and verify new autophagy pathways for the development of cancer.

## Results

**An integrative pipeline for the analysis of cancer-associated LIRCPs.** In this study, we develop a new pipeline named iCAL to form a link between aberrant autophagy selectivity and human cancer (Fig. 1). First, we design a sequence-based tool named pLIRm for predicting canonical LIR (cLIR) motifs that follow the sequence pattern [FWY]XX[LIV][5,6,16] (Fig. 1). A previously developed group-based prediction system (GPS) 5.0 algorithm has been considerably improved to measure the peptide similarity, and two additional approaches, including position weight determination and scoring matrix optimization, are implemented for performance improvement[17]. A widely used machine-learning algorithm, penalized logistic regression[18], is adopted for model training and parameter optimization (Fig. 1). Then, we map publicly available cancer mutations to human proteins and use pLIRm to score cLIR motifs without (Original) or with mutations (Mutant). We hypothesize that most cancer mutations located around cLIRs might exhibit weak influence, and we develop a model-based algorithm named pLAM to predict potential LAMs that significantly increase (Type I) or decrease (Type II) their binding potentials to LC3, using the Parzen window method (Eq. 13)[18]. Then, a pan-cancer analysis is conducted to analyze potential associations between LAM-containing LIRCPs and 37 major cancer types/subtypes (Fig. 1).

From the predicted LAM-containing LIRCPs, we select five proteins to test their interactions with LC3 and autophagy activities (Fig. 1). Among them, we focus on STBD1, a protein implicated in glycogen autophagy (glycophagy) but poorly characterized in other aspects. We use a combination of transcriptomics, metabolomics and additional experimental assays to study the role of STBD1 in tumor proliferation and the underlying mechanism. We envision that our pipeline will be useful to discover additional tumorigenesis pathways through the misregulation of autophagy selectivity.

**Sequence- and model-based prediction of cancer mutations that alter cLIR motifs.** From the literature, we manually collect 127 experimentally identified LIR motifs in 105 LIRCPs, including 89 and 11 LIR motifs in *Homo sapiens* and *Saccharomyces cerevisiae*, respectively (Fig. 2a, and Supplementary Data 1). Our benchmark data set is much larger than iLIR[19] and hfAIM[20], which only collected 27 and 36 known LIR motifs, respectively (Fig. 2a). We use a sequence logo generator WebLogo (http://weblogo.berkeley.edu/logo.cgi) to analyze the known LIR motifs, and find that F/W/Y and L/I/V residues are highly informative at positions 0 and +3 (Fig. 2b), which are consistent with the cLIR motif [FWY]XX[LIV][5,6,16]. Then, we use this data set for model training and develop a new tool named pLIRm. We compare pLIRm to other existing methods, including iLIR[19] and hfAIM[20], 3 reported LIR motifs and 4 sequence patterns in the eukaryotic linear motif database[21]. The leave-one-out validation and 4-, 6-, 8-, and 10-fold cross-validations are performed for pLIRm (Supplementary Data 2), whereas the accuracy values of other tools are directly calculated using the same benchmark data set. By comparison, pLIRm has a much higher area under the curve

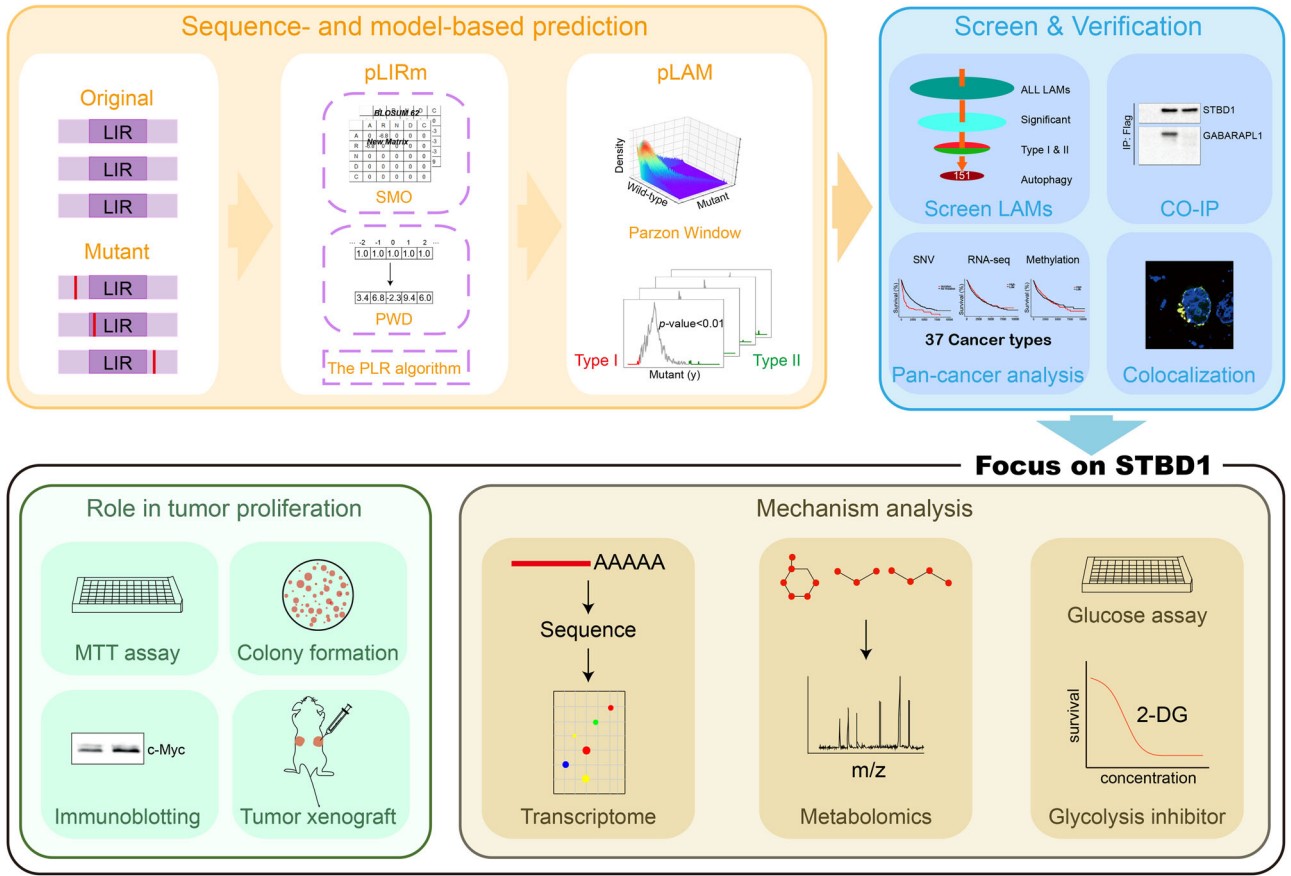

**Fig. 1 Major steps of iCAL.** i Design a sequence-based predictor, pLIRm, and develop a model-based approach, pLAM; ii computational prioritization of potential LAMs that significantly influence cLIR motifs, and then pan-cancer analysis and experimental validation of predicted LAM-containing LIRCPs; iii combine transcriptomics, metabolomics with additional experimental assays to study the role and mechanism of STBD1 in tumor proliferation; Co-IP co-immunoprecipitation.

(AUC) value than iLIR (0.8797 vs. 0.7810), which is better than or comparable with other previous methods (Fig. 2c). Thus, pLIRm is more accurate than other existing methods. More details on the comparison of pLIRm and other existing methods are also present (Supplementary Note 1, Supplementary Fig. 1).

From The Cancer Genome Atlas (TCGA)[22], International Cancer Genome Consortium (ICGC)[23] and Catalogue of Somatic Mutations in Cancer (COSMIC)[24], we obtain 2,963,952 non-redundant missense single nucleotide variants (SNVs). We map these cancer mutations to potential human LIRCPs predicted by pLIRm and identify 842,789 potential LAMs located in or around 238,840 cLIRs of 18,806 human proteins (Fig. 2d). Then, we develop a model-based algorithm named pLAM to prioritize LAMs that significantly change the binding potentials of cLIR motifs to LC3. For each LAM, the original and mutant peptides are pairwisely scored by pLIRm, with normalized values of $x$ and $y$, respectively (Fig. 2e). We use the Parzen window method (Eq. 13)[18] to estimate the global distribution of $x$ and $y$, and calculate the significance of $y$ values under a given $x$ score. Under a threshold of $p$ value < 0.01, Type I and II LAMs are identified based on the mutated score $y > 0.5$ and the original score $x > 0.5$, respectively (Fig. 2e). Finally, reserved LAMs are mapped to known ATG proteins and autophagy regulators. In total, we identify 222 potential LAMs including 60 Type I and 162 Type II LAMs that significantly change 172 cLIR motifs in 148 LIRCPs (Fig. 2d and Supplementary Data 3).

Using the hypergeometric test, the enrichment analyses are performed for the 148 potential LIRCPs based on the annotations of gene ontology (GO) biological processes (Fig. 2f, $p$ value < 10$^{-5}$)

and Kyoto Encyclopedia of Genes and Genomes (KEGG) pathways (Fig. 2g, $p$ value < 10$^{-7}$). The GO-based results demonstrate that core autophagy processes are significantly over-represented, indicating that ATG proteins and autophagy regulators have been truly enriched in the finally-prioritized LIRCPs (Fig. 2f). KEGG-based analysis reveals several enriched cancer-associated pathways, indicating a strong correlation between autophagy and human cancer (Fig. 2g).

**LAM-containing LIRCPs play a potential role in human cancer.** To analyze the associations of the 148 predicted LIRCPs in human cancer, we download TCGA data sets including cancer single nucleotide variants (SNVs), RNA sequencing (RNA-seq), and DNA methylation profiles, as well as corresponding clinical outcomes of 37 major cancer types/subtypes[22]. Survival analyses of the association between the TCGA data and clinical outcomes are performed for each layer of the omics data in both pan-cancer and individual cancer levels (Supplementary Data 4, two-sided log-rank test, SNV: $p$ value < 0.05; RNA expression: $p$ value < 10$^{-4}$; DNA methylation: $p$ value < 10$^{-4}$). The pan-cancer analysis reveals that SNVs, RNA expressions, and DNA methylation levels of 18, 100, and 108 LIRCPs are statistically associated with human cancer (Fig. 3a). For individual cancer types, the results of RNA-seq-based survival analyses for several LIRCPs are shown (Fig. 3a). It can be found that the gene expression levels of a number of ATG proteins, such as ATG2B, ATG4A, ATG5, ATG9A, and SNX4/SNX30 (Orthologs of Atg20/Atg24 in *S. cerevisiae*), are associated with the survival rate in multiple cancer types (Fig. 3b).

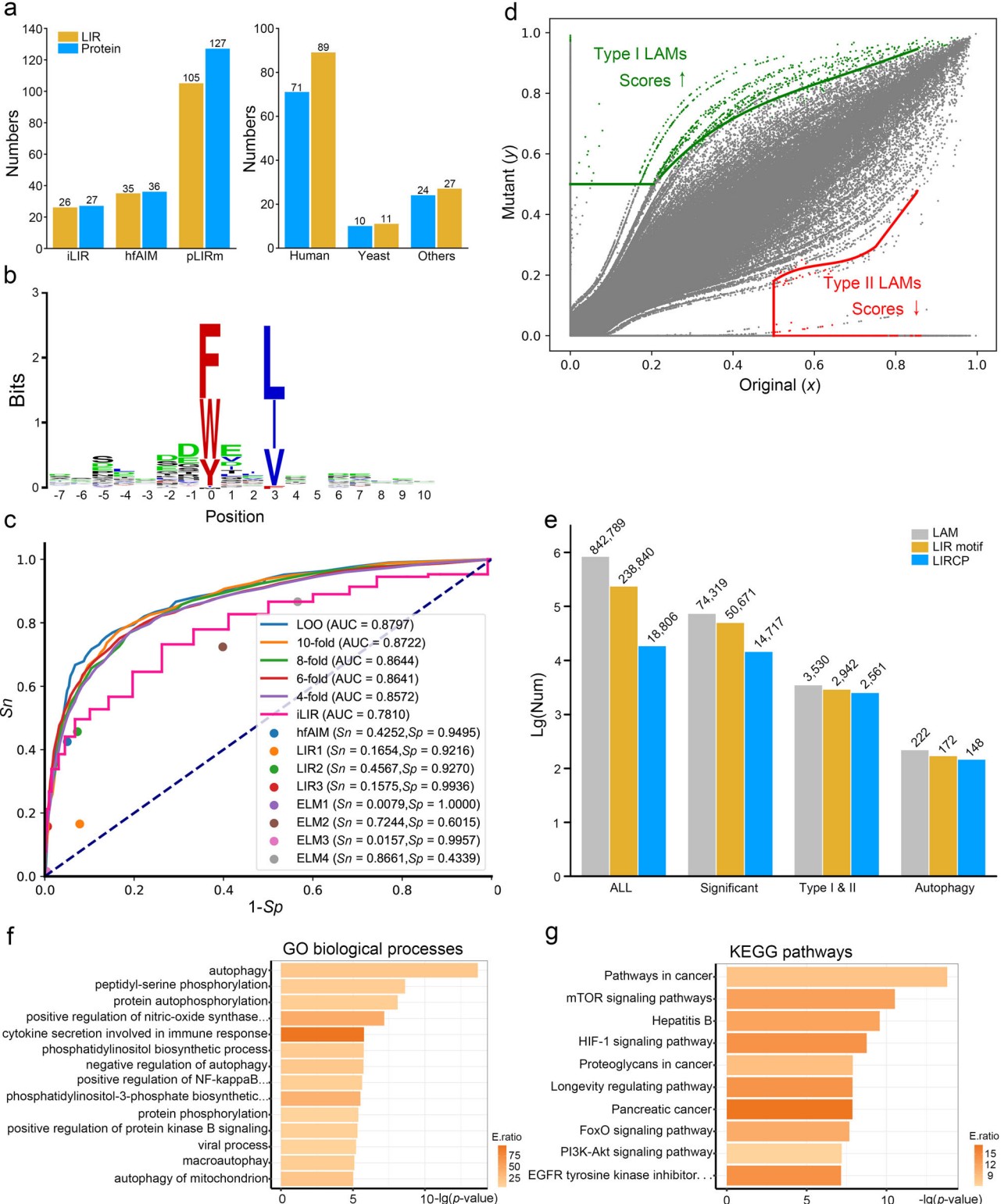

**Fig. 2 Computational prioritization of highly potential LAM-containing LIRCPs. a** A comparison of known LIR motifs and corresponding proteins collected by iLIR[19], hfAIM[20], and pLIRm, as well as the distribution of our collected data in *H. sapiens* and *S. cerevisiae* and other species (Supplementary Data 1). **b** A sequence logo of known LIR motifs was generated by WebLogo (http://weblogo.berkeley.edu/logo.cgi)[76]. **c** A comparison of pLIRm to other methods, including iLIR[19], hfAIM[20], three LIR motifs (WXXL, [ADEFGLPRSK][DEGMSTV][WFY][DEILQTV][ADEFHIKLMPSTV][ILV], and [DE][DEST][WFY][DELIV]x[ILV])[5,19,77], and four ELM motifs ([EDST].{0,2}[WFY]..P, [EDST].{0,2}[WFY][^RKPG][^PG][ILV], [EDST].{0,2}LVV, and [EDST].{0,2}[WFY]..[ILVFY])[21]. **d** The model-based algorithm pLAM for predicting Type I and Type II LAMs that potentially increase and decrease the binding affinity of cLIR motifs to LC3, respectively. **e** The distribution of numbers of potential LAMs, LIR motifs and LIRCPs reserved in each step of pLAM. **f**, **g** The GO- and KEGG-based enrichment analyses of finally reserved LAM-containing LIRCPs.

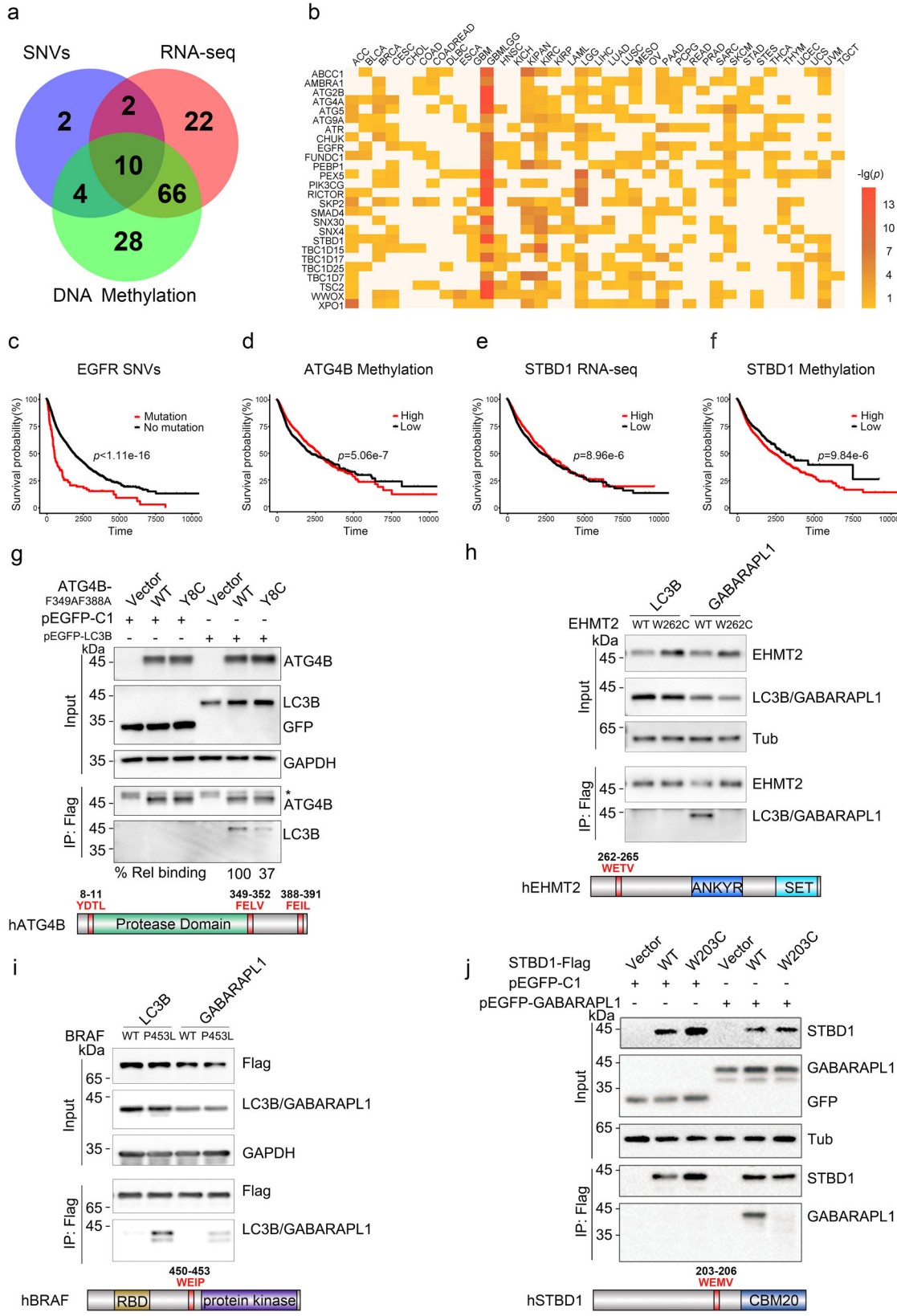

From the pan-cancer analysis, we find ten genes to be associated with cancer in all three omics layers. For example, it can be clearly found that SNVs in EGFR, a well-characterized oncogenic protein tyrosine kinase, is highly associated with a lower survival probability in pan-cancer (Fig. 3c). Although SNVs and RNA expression levels of ATG4B are not detected to be associated with cancer, the lower

DNA methylation level of ATG4B is statistically correlated with a higher survival rate, supporting its oncogenic role in tumorigenesis (Fig. 3d). In particular, we observe STBD1, a protein involved in glycophagy, to be potentially associated with human cancer (Supplementary Data 4). At the pan-cancer level, it is found that a higher mRNA expression level or a lower DNA methylation level

**Fig. 3 LAMs in ATG4B, EHMT2, BRAF, and STBD1 found in cancer patients impair their interactions with ATG8. a** The association of LAM-containing LIRCPs and human cancer at SNV, RNA-seq, and DNA methylation levels. **b** The expression levels of LAM-containing ATG proteins and autophagy regulators across different cancer types. **c** Mutated EGFR is associated with a shorter survival rate. **d** Low DNA methylation level of ATG4B is associated with a longer survival rate. **e** High mRNA expression level of STBD1 is associated with a longer survival rate. **f** Low DNA methylation level of STBD1 is associated with a longer survival rate. In **c–f**, significance (*p* value) is determined by a two-sided log-rank test. **g** HEK293T cells were co-transfected with Flag-tagged ATG4B wild-type (WT) or Y8C and GFP-tagged LC3B for 24 h. All ATG4B plasmids were made in the F349A/F388A background to minimize the roles of LIR2 and LIR3. Control cells were transfected with an empty vector. One-tenth of the cell lysate was prepared as input, and the rest was used for immunoprecipitation with anti-Flag Sepharose 4B gel followed by immunoblotting with indicated antibodies. The band of LC3B was quantified by Image J and normalized to the level of ATG4B WT, and labeled below the blots. Schematic representation of human ATG4B with red fonts indicating the LIR motif. *IgG heavy chain. **h, i** HEK293T cells were co-transfected with Flag-tagged EHMT2 (160–360 aa) or BRAF and GFP-tagged LC3B or GABARAPL1 for 24 h. Control cells (vector) were transfected with an empty vector. One-tenth of the cell lysate was prepared as input, and the rest was used for immunoprecipitation (IP) with anti-Flag Sepharose 4B gel followed by immunoblotting with indicated antibodies. Schematic representation of human EHMT2 and BRAF with red fonts indicating the LIR motif. **j** HEK293T cells were co-transfected with Flag-tagged STBD1 WT or W203C and GFP-tagged GABARAPL1 for 24 h. Control cells (vector) were transfected with an empty vector. One-tenth of the cell lysate was prepared as input, and the rest was used for immunoprecipitation with anti-Flag Sepharose 4B gel followed by immunoblotting with indicated antibodies. Schematic representation of human STBD1 with red fonts indicating the LIR motif. Experiments **g–j** were performed in triplicate.

of STBD1 is significantly associated with a higher survival probability (Fig. 3e, f, Supplementary Data 4), indicating STBD1 might, in general, have tumor-suppressive functions. However, in the individual cancer level, we find that the higher mRNA expression level in glioma (GBMLGG) and lower DNA methylation level in GBMLGG and brain lower-grade glioma (LGG) of STBD1 are significantly associated with a lower survival probability, exhibiting an opposite result against that in the pan-cancer level (Supplementary Data 4, Supplementary Fig. 2). Thus, STBD1 might have different roles in distinct types of cancer.

**LAMs in ATG4B, EHMT2, BRAF, and STBD1 impair their interactions with ATG8.** To test our predictions, we begin to probe the interactions of ATG4B with LC3B. ATG4B has three putative LIR motifs: LIR1 ($_8$YDTL$_{11}$) at the N-terminus, LIR2 ($_{349}$FELV$_{352}$) just C-terminal to the protease domain, and LIR3 localized within the C-terminus of the protein ($_{388}$FEIL$_{391}$)[25] (Fig. 3g). A Type II (decrease binding) mutant found in cancer samples is within the N-terminal LIR motif (Y8C). ATG4B Y8C shows decreased binding to LC3B, with the bound LC3B being 37% of wild type (Fig. 3g). Overexpression of ATG4B Y8C or enzymatically inactive mutant (C74S) impairs LC3B lipidation, as defined by the ratio of LC3B II to LC3B I (Supplementary Fig. 3a). These results collectively demonstrate that ATG4B cancer mutation can diminish its LC3B binding and autophagy activities.

To further assess whether our algorithm is useful to predict new LIR motifs, we select three proteins that have no reported LIR motifs: EHMT2, a histone methyltransferase; ERCC6, a protein involved in DNA repair; and BRAF, a serine/threonine–protein kinase (Supplementary Data 5). Among them, our algorithm predicts that EHMT2 and ERCC6 have potential LIR motifs (EHMT2: $_{262}$WETV$_{265}$; ERCC6: $_{1282}$VEAE$_{1285}$), which are disrupted by cancer-associated mutations. In contrast, a Type I mutation (P453L) in BRAF is predicted to gain interaction with LC3 (Supplementary Data 5). Indeed, the co-immunoprecipitation experiment reveals that EHMT2 WT can specifically interact with GABARAPL1, and the association is disrupted by the W262C mutation (Fig. 3h). Notably, the BRAF P453L mutation shows an increased binding affinity with LC3B and GABARAPL1 relative to BRAF WT (Fig. 3i). Finally, we do not detect the interaction between ERCC6 with LC3B or GABARAPL1 (Supplementary Fig. 3b). Taken together, our experimental data demonstrate that our algorithm is helpful in detecting known LIR motifs as well as predicting new motifs.

STBD1 is proposed to be an adaptor protein for glycogen autophagy, but otherwise poorly studied[26]. STBD1 encompasses

an LIR sequence ($_{203}$WEMV$_{206}$) that interacts with GABARAPL1[27], and a Type II mutant in the LIR sequence (W203C) is derived from one of the 97 intestinal adenocarcinoma samples in the COSMIC database[24] (Fig. 3j, Supplementary Data 3). This mutation has been annotated as "Pathogenic" with a score of 0.97 predicted by FATHMM, a Hidden Markov Model-based web-server to predict the functional impacts of both coding and noncoding variants in the human genome[28]. Indeed, whereas STBD1 WT robustly immunoprecipitates GABARAPL1, the W203C mutant completely abolishes the interaction (Fig. 3j). Co-localization analysis reveals that STBD1 WT strongly co-localizes with GABARAPL1, and W203C disrupts the co-localization (Supplementary Fig. 3c). Accordingly, overexpression of Flag-tagged STBD1 WT, but not the W203C mutant, results in an increased ratio of LC3B II/LC3B I (Supplementary Fig. 3d). Furthermore, shRNA knockdown STBD1 decreases the ratio of LC3B II/LC3B I, which can be rescued by the over-expression of STBD1 WT, but not by that of STBD1 W203C (Supplementary Fig. 3e). Overexpression of STBD1 WT, but not W203C, induces degradation of p62, in the absence of BafA1 (Supplementary Fig. 3f). The presence of BafA1, however, leads to the accumulation of p62 in both cells (Supplementary Fig. 3f). These data indicate that cancer-associated mutation of STBD1 on W203 abrogates its binding to GABARAPL1 and impairs its functions in autophagy.

**STBD1 inhibits tumor growth in multiple cancer cells and in vivo.** We next focus on STBD1, a proposed mediator of glycophagy. Although the connection between autophagy and human cancer is well-established, the functions of glycophagy in cancer development are currently unknown. To investigate whether the expression of STBD1 is altered in tumor samples, we performed immunohistochemistry (IHC) staining to examine 27 colon cancer specimens and paired adjacent noncancerous tissues. The expression of STBD1 at the protein level is significantly lower in the cancer tissues in comparison with that in the adjacent non-carcinoma tissues (~56% of paracancer) (Supplementary Fig. 4a). In cells expressing STBD1 WT, glycogen strongly co-localizes with GABARAPL1, and the co-localization is nearly abolished in cells expressing STBD1 W203C (Fig. 4a). Furthermore, overexpression of STBD1 W203C, but not STBD1 WT, increases the total glycogen levels in HCT116 cells, indicating that STBD1 promotes glycogen metabolism via associating with GABARAPL1 (Fig. 4b). Interestingly, overexpression of STBD1 WT, but not STBD1 W203C, significantly suppresses cell growth and colony formation in lung cancer cell line A549 cells (Fig. 4c–e, colony formation: plvx neo = 775 ± 30; STBD1

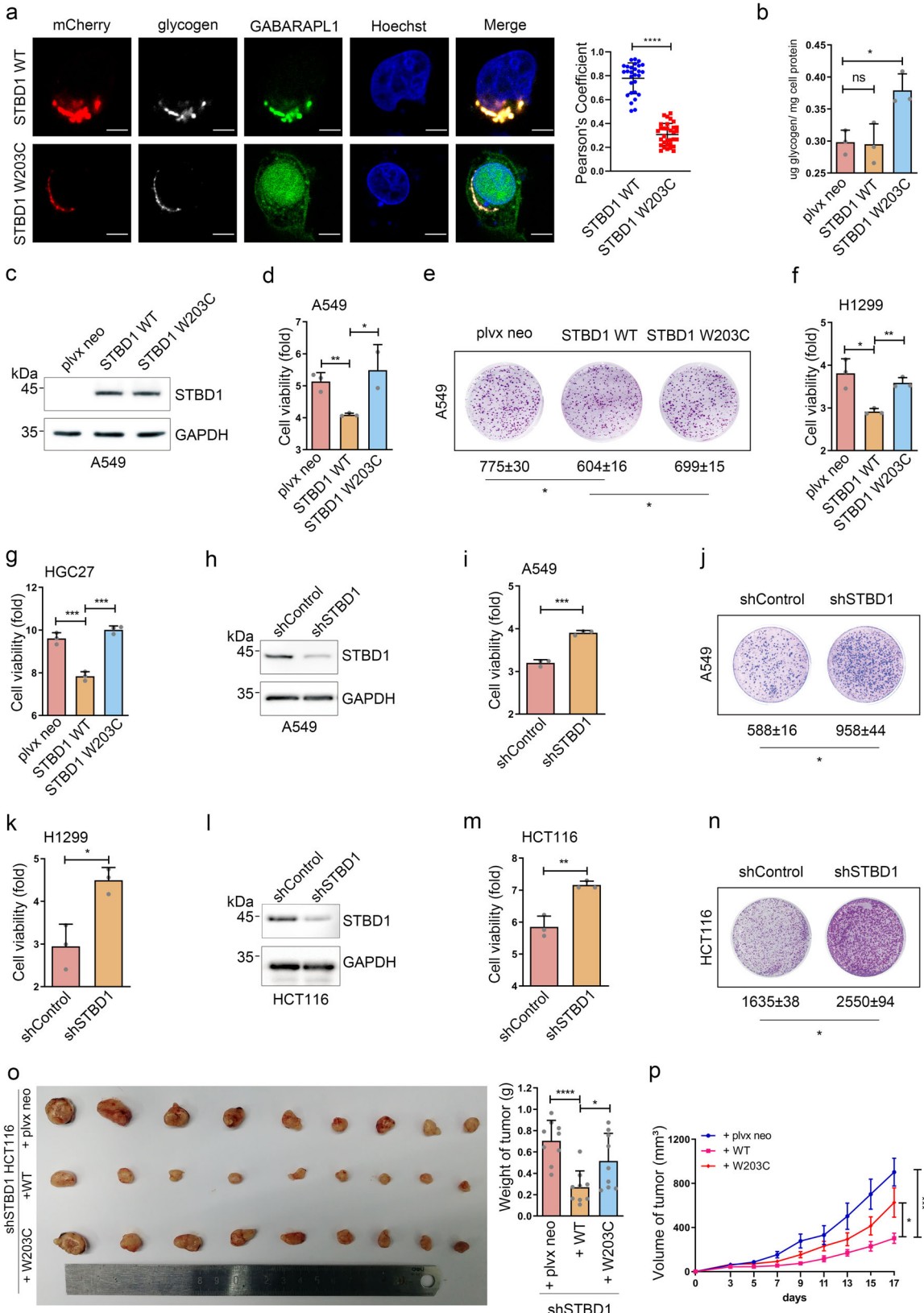

WT = 604 ± 16; STBD1 W203C = 699 ± 15). Similar results are obtained in another lung cancer cell line, H1299 cells (Fig. 4f, and Supplementary Fig. 4b, c).

To determine whether STBD1 plays a similar role in other types of cancer cells, we survey the expression of STBD1 in different cell lines and find that gastric cancer cell line HGC27 has the lowest expression (Supplementary Fig. 4d). In order to minimize the potential effect of endogenous STBD1, we thus choose HGC27 cells for the following studies. Similar to our results from lung cancer cell lines, over-expression of STBD1 WT,

**Fig. 4 STBD1 inhibits tumor growth in vitro and in vivo. a** Confocal immunofluorescence of HeLa cells co-transfected with mCherry-STBD1 WT or W203C, and GFP-tagged GABARAPL1 for 24 h. Glycogen was stained anti-glycogen monoclonal antibody IV58B6 (white) with Nuclei was stained with Hoechst (blue). Images were captured using the Olympus FV-1000. Pearson's coefficients of glycogen and GABARAPL1 were calculated using image J. Each dot represents the value of one cell. Scale bar, 20 μm. **b** The glycogen content of HCT116 cells stably expressing plvx neo, STBD1 WT or STBD1 W203C, respectively, was assessed using a glycogen assay kit. **c** Control A549 cells (plvx neo) and A549 cells stably overexpressing STBD1 WT or W203C were lysed for immunoblotting to determine the protein levels of STBD1 and GAPDH. **d** Control A549 cells (plvx neo) and A549 cells stably overexpressing STBD1 WT or STBD1 W203C were cultured for 72 h. The cell viability was assessed using the MTT assay and normalized to that of 0 h. **e** Control A549 cells (plvx neo) and A549 cells stably overexpressing STBD1 WT or STBD1 W203C were cultured for 20 days, then stained by crystal violet. The number of colonies was analyzed using Image J. **f, g** Control H1299/HGC27 cells (plvx neo) and H1299/HGC27 cells stably expressing STBD1 WT or W203C were cultured for 72 h. Cell viability was then assessed using the MTT assay and normalized to that of 0 h. **h** The protein levels of STBD1 in shControl (control shRNA) and shSTBD1 A549 cells were determined by immunoblotting. **i** shControl (control shRNA) and shSTBD1 A549 cells were cultured for 72 h. The cell viability was then assessed using the MTT assay and normalized to that of 0 h. **j** shControl and shSTBD1 A549 cells were cultured for 20 days, then stained by crystal violet. The number of colonies was analyzed using Image J. **k** shControl and shSTBD1 H1299 cells were cultured for 72 h. The cell viability was then assessed using the MTT assay and normalized to that of 0 h. **l** The protein levels of STBD1 in shControl and shSTBD1 HCT116 cells were determined by immunoblotting. **m** shControl and shSTBD1 HCT116 cells were cultured for 72 h. The cell viability was then assessed using the MTT assay and normalized to that of 0 h. **n** shControl and shSTBD1 HCT116 cells were cultured for 20 days, then stained by crystal violet. The number of colonies was analyzed using Image J. **o** Nude mice ($n = 9$) were injected subcutaneously on the back of the neck or both flanks with shSTBD1/plvx neo, shSTBD1/WT, or shSTBD1/W203C HCT116 cells, respectively. Images show the dissected tumors and tumor weights 17 days after injection. **p** Tumor volume was measured over time after injection in mice as in (**o**). Experiments in **a–n** were performed in triplicate. **a–p** Statistical data are presented as mean ± SD. Statistical comparisons were performed using an unpaired t test. ****$p < 0.0001$, ***$p < 0.001$, **$p < 0.01$, *$p < 0.05$.

but not STBD1 W203C, in HGC27 cells, reduces the cell proliferation rate (Fig. 4g, and Supplementary Fig. 4e, f). Thus, STBD1 inhibits cell proliferation in multiple types of cancer cells.

To further confirm our observations, we test how the depletion of STBD1 affects cancer growth. Using shRNA especially targeting STBD1 (shSTBD1), we are able to effectively suppress the expression of STBD1 in A549 cells (Fig. 4h). These cells grow significantly faster, and form more colonies, relative to control cells (Fig. 4i, j, colony formation: shControl = 588 ± 16; shSTBD1 = 958 ± 44). Silencing of STBD1 in H1299 cells also yields similar results (Fig. 4k, and Supplementary Fig. 4g, h). To further confirm our results using shSTBD1 cells, we generate two STBD1 knockout-A549 clonal cell lines using the CRISPR–Cas9 technology (Supplementary Fig. 4i). These two cell lines display a higher proliferation rate (Supplementary Fig. 4j). Mutation of STBD1 (W203C) is found in patients with intestinal adenocarcinoma (Supplementary Data 5), and we also examined colorectal carcinoma cell line HCT116. As shown in Fig. 4l–n, knockdown of STBD1 in HCT116 cells promotes cell growth and colony formation (colony formation: shControl = 1635 ± 38; shSTBD1 = 2550 ± 94).

To test whether STBD1 inhibits cancer growth in vivo, we establish a tumor xenograft model through subcutaneous injecting HCT116 cells into immunodeficient mice. Depletion of STBD1 results in enhanced tumor growth in mice (Supplementary Fig. 5a, b; the weight of tumor: shControl = 0.44 ± 0.08; shSTBD1 = 0.7539 ± 0.10). Confirming this observation, immunohistochemical analysis reveals that the proliferation biomarker Ki67 is significantly higher in shSTBD1 tumors compared to shControl tumors (Supplementary Fig. 5c, mean ± SD, shControl = 100,708 ± 16,735; shSTBD1 = 115,212 ± 17,070). In addition, the TUNEL assay shows that knockdown of STBD1 has no effect on the cell apoptosis (Supplementary Fig. 5d, mean ± SD, shControl = 2.530 ± 0.911; shSTBD1 = 2.476 ± 0.948).

To investigate whether STBD1 suppresses tumor growth via glycophagy, the shSTBD1 HCT116 cells are overexpressed with shRNA-resistant STBD1 WT (shSTBD1/WT) and W203C (shSTBD1/W203C). In these cells, overexpression of STBD1 WT, but not STBD1 W203C, significantly decreases cell growth (Supplementary Fig. 6a, b). Furthermore, shSTBD1/WT markedly suppresses tumor growth in the xenograft model, in comparison with control (plvx neo) and shSTBD1/W203C (Fig. 4o, p; the weight of tumor: plvx neo = 0.704 ± 0.181; STBD1 WT = 0.269 ± 0.145;

STBD1 W203C = 0.515 ± 0.244). IHC analysis further reveals that the proliferation biomarker Ki67 decreases in shSTBD1/WT samples, in comparison with control and shSTBD1/W203C (Supplementary Fig. 6c, mean ± SD, plvx neo = 56,691 ± 13,593; STBD1 WT = 48,967 ± 11,550; STBD1 W203C = 53,468 ± 7611). Overexpression of STBD1 WT or W203C, however, does not drastically affect cell apoptosis (Supplementary Fig. 6d). Taken together, our results indicate that STBD1 has potential tumor-suppressive activity through interacting with LC3B and participating in glycophagy.

To further determine whether STBD1 regulates tumor growth via a role in glycophagy, we test another mediator of glycophagy, lysosomal acid α-acid glycosidase (GAA). When glycogen is delivered to the lysosomal compartment, it is subsequently degraded via GAA[29]. Using two different siRNAs especially targeting GAA, we are able to achieve the knockdown efficiency of 50–90%, measured by quantitative real-time polymerase chain reaction (qRT-PCR) (Supplementary Fig. 7a, b). Similar to STBD1, each of the two GAA siRNA-transfected HCT116 and A549 cells grow significantly faster, relative to that of the siControl cells (Supplementary Fig. 7c, d). Furthermore, GAA depletion also promotes the formation of colonies (Supplementary Fig. 7e, f, colony formation, HCT116: siControl = 176 ± 7; siGAA-1 = 235 ± 10; siGAA-2 = 237 ± 7; A549: siControl = 235 ± 11; siGAA-1 = 356; siGAA-2 = 297 ± 10). Altogether, our studies discover previously uncharacterized roles of glycophagy in tumor growth and suggest that aberrant expression or mutations of related genes could contribute to the pathogenesis of cancer.

**STBD1 mutation or depletion favors the acquisition of multiple cancer hallmark traits.** To understand how STBD1 inhibits tumor growth, we compare the gene expression profiles of shControl and shSTBD1 HCT116 cells using RNA-seq (Supplementary Data 6). Three biological replicates of shControl and shSTBD1 are separately clustered, and 454 genes are differentially expressed, including 263 upregulated and 191 downregulated genes ($p$ value $< 10^{-5}$) (Fig. 5a–d). KEGG-based enrichment analysis indicates that these differentially expressed genes (DEGs) are enriched in cancer-related pathways, such as transcriptional misregulation and microRNAs in cancer, indicating a potential role of STBD1 in cancer development. Glycolysis/gluconeogenesis (KEGG: hsa00010) is another enriched process, consistent with the role of STBD1 as a cargo receptor for glycogen (Fig. 5d).

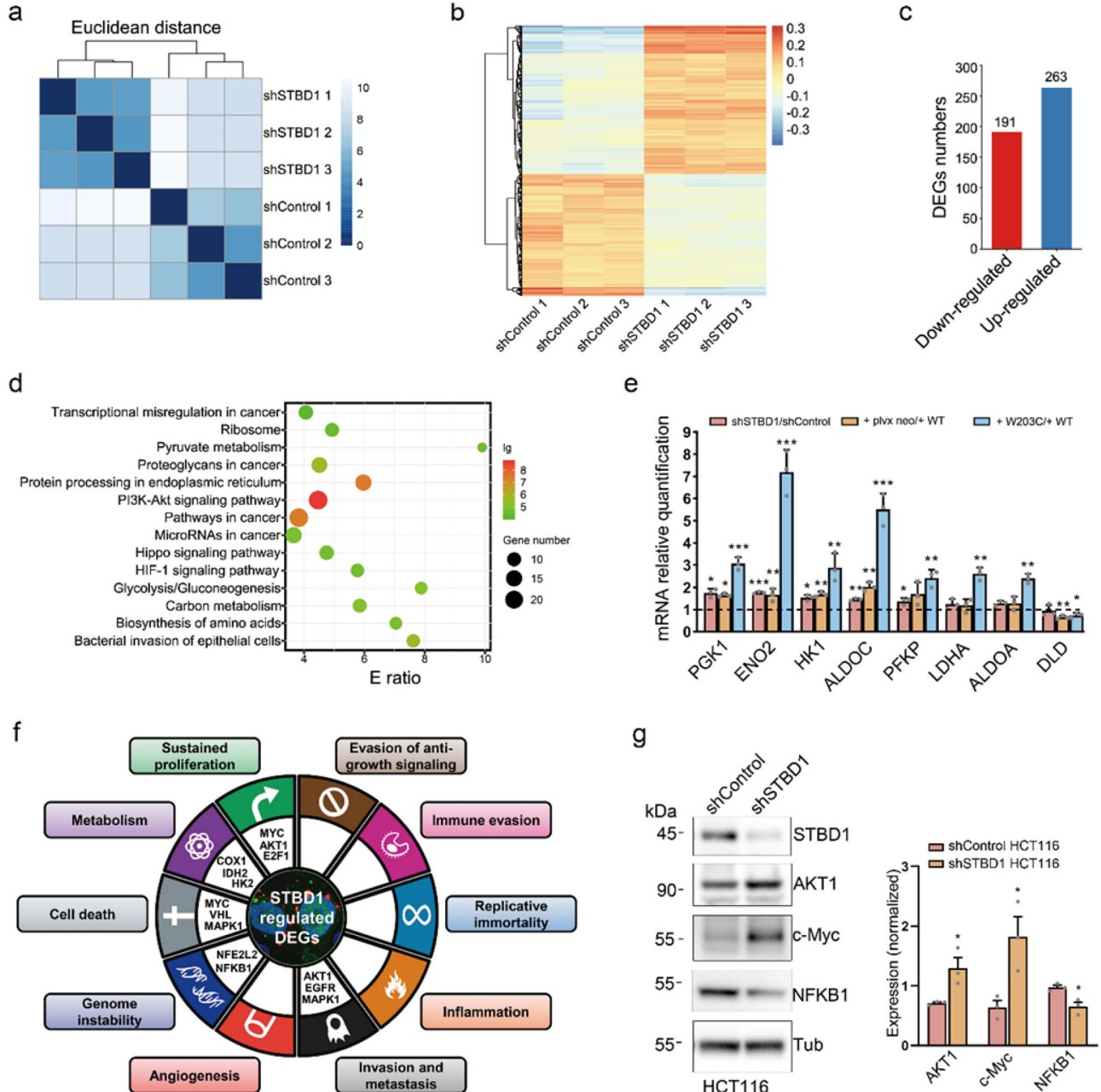

**Fig. 5 Depletion of STBD1 alters multiple genes critical for glycolysis. a** Three biological replicates of shControl and shSTBD1 HCT116 cells were clustered together in RNA-seq. **b** Heatmap of differentially expressed genes in shControl and shSTBD1 HCT116 cells. **c** The numbers of down-regulated or up-regulated genes in differentially expressed genes. **d** The KEGG-based enrichment analysis of biological pathway of differentially expressed genes. **e** mRNA levels of several genes responsible for glycolysis were determined by RT-PCR and normalized using TUBB mRNA. These relative mRNA levels in shSTBD1 vs. shControl HCT116 cells, shSTBD1/W203C vs. shSTBD1/WT cells, and shSTBD1/WT vs. shSTBD1/plvx neo cells were shown. **f** Enrichment analysis of cancer hallmark traits affected by STBD1 depletion. Representative hallmark genes are shown in the circle. **g** Protein levels of STBD1, AKT1, c-Myc, and NFKB1 (p50) in shControl and shSTBD1 HCT116 cells were determined by immunoblotting and normalized using Tubulin or GAPDH. Experiments in **e**, **g** were performed in triplicate. Statistical data are presented as mean ± SD. Statistical comparisons were performed using an unpaired *t* test. ***$p < 0.001$, **$p < 0.01$, *$p < 0.05$.

To further confirm our observations, we also compare the gene expression profiles of shSTBD1/plvx neo, shSTBD1/WT, and shSTBD1/W203C cells using RNA-seq (Supplementary Data 7). Remarkably, 8 out of 14 pathways, including glycolysis/gluconeogenesis, enriched in the shSTBD1 cells are also found in the ones that are differentially affected in shSTBD1/WT and shSTBD1/W203C cells (Supplementary Fig. 8a–d). To further validate these results, we perform quantitative RT-PCR and compare the

expression of multiple DEGs responsible for glycolysis/gluconeo-genesis in three different groups: (1) shSTBD1 vs. shControl, (2) shSTBD1/plvx neo vs. shSTBD1/WT, (3) and shSTBD1/W203C vs. shSTBD1/WT cells (Fig. 5e). The three sets of comparisons reveal a highly similar pattern, suggesting that STBD1 W203C mutation leads to the loss of normal functions of STBD1, similar to shSTBD1. Seven out of eight genes that we have examined, including PGK1, ENO2, ALDOC, HK1, PFKP, LDHA, and

ALDOA, are elevated in at least one set of comparisons, consistent with our RNA-seq results (Fig. 5e). The eighth gene, DLD, is modestly decreased in two groups, shSTBD1/plvx neo vs. shSTBD1/WT, and shSTBD1/W203C vs. shSTBD1/WT cells (Fig. 5e). Overall, our data suggest that STBD1 inhibits cancer growth, likely through altering gene transcription and rewiring the glycolysis/gluconeogenesis pathway.

We obtain 377 known cancer hallmark genes from a well-curated database named HOC[30] and map them to our transcriptomic data (Supplementary Data 6). For the 1611 DEGs with a relaxed stringency ($p$ value < 0.01), enrichment analyses demonstrate that five cancer hallmark traits, including sustained proliferation, genome instability, cell death, invasion and metastasis, and metabolism, are markedly affected upon STBD1 depletion (Fig. 5f, $p$ value < 0.05). Two hallmark traits, sustained proliferation, and cell death are validated to be affected by STBD1, through immunoblotting of c-Myc (Fig. 5g). The c-Myc expression level is found to be significantly upregulated in shSTBD1 cells (Fig. 5g). In addition, we examine the expression of two hallmark genes: NFKB1 and AKT1. Indeed, STBD1 depletion leads to up-regulation of oncogene AKT1, whereas suppresses the expression of tumor suppressor NFKB1 (Fig. 5g), indicating that other pro-tumorigenic pathways are also upregulated. Consistently, both IHC and immunoblotting analysis reveal that the c-Myc level is significantly higher in shSTBD1 tumors relative to shControl tumors (Supplementary Fig. 9a, b, c-Myc: shControl = 91,173 ± 23,991; shSTBD1 = 147,159 ± 47,662). Furthermore, STBD1 WT, but not W203C, suppresses the expression of c-Myc (Supplementary Fig. 9c, d, c-Myc: plvx neo = 72,789 ± 19,915; STBD1 WT = 62,204 ± 16,410; STBD1 W203C = 69,981 ± 18,519). Taken together, our results suggest that STBD1 suppresses tumor growth by inhibiting multiple cancer hallmark traits.

**STBD1 depletion promotes glycolysis in cancer cells.** The above findings suggest that STBD1 depletion potentially leads to metabolic reprogramming. To probe such changes, we first perform targeted metabolomic profiling of shControl and shSTBD1 HCT116 cells, each with three biological replicates (Fig. 6a, Supplementary Data 8). More than 200 metabolites in various metabolic pathways, including glycolysis, tricarboxylic acid (TCA) cycle, purine metabolism, pyrimidine metabolism, and amino acids, are observed (Fig. 6b, Supplementary Data 8). To trace the glycolytic flow in cancer cells, we further perform an isotope tracing analysis using stable $^{13}C_6$-glucose labeling (Fig. 6c–f, Supplementary Data 8). Knockdown of STBD1 leads to increased glycolytic intermediates, as represented by 3-Phosphoglycerate/2-Phosphoglycerate (3-PG/2-PG) $m + 3$, phosphoenolpyruvate (PEP) $m + 3$, and pyruvate $m + 3$ in the glycolysis pathway (Fig. 6c). Meanwhile, enhanced glucose metabolism into TCA cycle, e.g., citrate $m + 2$, aconitate $m + 2$, isocitrate $m + 2$, and α-ketoglutarate (α-KG) (Fig. 6d), is observed, as well as nucleotide biosynthesis through pentose phosphate pathway, e.g., AMP $m + 5$ in purine metabolism and UMP $m + 5$ in pyrimidine metabolism (Fig. 6e, f). The unchanged intracellular level of lactate $m + 3$ (Fig. 6c) further confirms the enhancement of glucose metabolism is biased into oxidative phosphorylation and nucleotide biosynthesis, and the latter is required to make RNA and DNA in proliferating cells in shSTBD1 cells. The results are highly consistent with our observation in the transcriptomics (Fig. 6g). In contrast, most essential amino acids are not altered by knockdown of STBD1, based on the targeted metabolomic profiling (Supplementary Fig. 10a). Taken together, depletion of STBD1 leads to substantial reprogramming of glucose metabolism in cancer cells through enhanced glycolysis.

Analysis of the medium reveals that the glucose level is lower in shSTBD1 cells than that of shControl cells, whereas no significant

difference of lactate level is observed (Fig. 6h). Conversely, overexpression of STBD1 WT, but not W203C, increases the glucose level in the medium (Supplementary Fig. 10b). As deletion of STBD1 can enhance glycolysis in cancer cells, we speculate that shSTBD1 HCT116 is more dependent on exogenous glucose. Indeed, glucose starvation impairs the growth of shSTBD1 cells more significantly than shControl cells (Fig. 6i, and Supplementary Fig. 10c). Since shSTBD1 HCT116 cells are more dependent on glycolysis than shControl cells, depletion of STBD1 may sensitize cancer cells to glycolysis inhibition. To test this hypothesis, we treat cells with 2-deoxy-d-glucose (2-DG), a glucose analog that inhibits phosphorylation of glucose by hexokinase[31]. Whereas 2-DG inhibits the growth of shSTBD1 and shControl HCT116 cells, the proliferation of shSTBD1 cells is decreased more than shControl cells under various concentrations of 2-DG (Fig. 6j). These results are further confirmed by STBD1 knockout A549 cells, in which we find that both independent clones display more sensitivity than control cells (Supplementary Fig. 10d). Altogether, we discover that STBD1 has putative tumor-suppressive functions, and our findings indicate that mutation or lower expression of STBD1 may promote cancer growth in patients. Targeting glycolysis could represent a promising approach to treat these patients.

**A LIRCP-regulating network links autophagy selectivity and tumorigenesis.** Understanding the mechanisms whereby the autophagy network interfaces with cancer is a long-standing challenge. The central questions include whether the autophagy pathways are targets for recurring molecular alteration in human cancer, and which pathways are targeted[1,2]. To identify the autophagy pathways perturbed in human cancer, we model a LIRCP-regulating network by integrating protein–protein interactions (PPIs) and transcriptional regulations among the 148 identified LIRCPs, 7 LC3 proteins, and 14 proteins regulated by STBD1 since both mechanisms are important for regulating autophagy[32–35] (Fig. 7).

Based on the functional annotations in UniProt[36], we classify 148 LIRCPs into nine classes, including apoptosis-associated events, autophagic vacuole assembly, cell cycle/proliferation, small GTPase-associated signaling, inflammatory/immune response, metabolic pathways, PI3K/AKT/mTOR signaling, biomolecule/vesicle transport, and glycolysis. The seven LC3 proteins are categorized into the class of autophagic vacuole assembly, based on their important role in autophagosome formation. The 14 downstream proteins of STBD1 are also included. Known or predicted PPIs and transcriptional regulations between transcription factors and target genes are integrated from 8 public databases, including ARN[32], BioGrid[37], IID[38], inBio MapTM[39], Mentha[40], HINT[41], iRefIndex[42], and PINA[43]. In total, we obtain 2204 PPIs and 91 transcriptional regulations for the 169 proteins, in which known cancer hallmark proteins are also indicated[35] (Fig. 7). From the network, it can be found how LIRCPs affect human cancer through the nine functional aspects, and highlight the functional importance of STBD1 in inhibiting cancer growth through modulating glycophagy (Fig. 7). Our work indicates cancer cells frequently alter autophagy selectivity for survival.

**Discussion**

The importance of autophagy for cancer initiation and growth cannot be understated. Autophagy helps to maintain normal cell homeostasis by removing oncogenic substances, such as toxic unfolded proteins and damaged organelles[44]. Autophagy also plays important role in malignant transformation, tumor progression, and treatment response. Studies using human cancer samples have also revealed that multiple *ATG* genes and

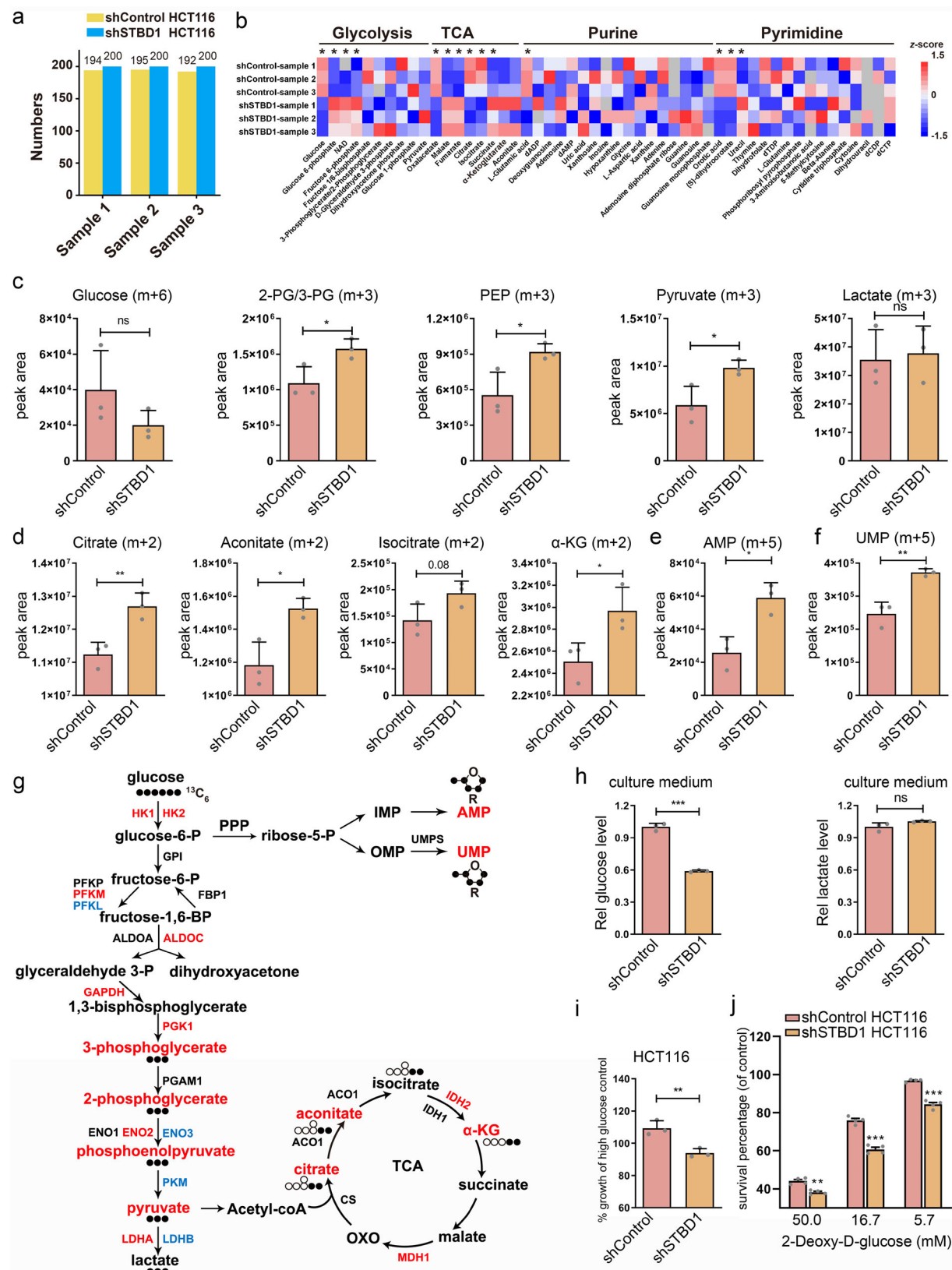

autophagy regulators aberrantly expressed or significantly mutated in human tumors[14,15]. However, prior to our work, it was unknown whether naturally occurring mutations exist in cancer samples that specifically alter autophagy selectivity. In this study, we implement a new pipeline named iCAL that integrates a sequence-based predictor, a model-based computational method,

publicly available cancer mutations, and multiple experimental approaches. This pipeline allows us to discover 222 LAMs in 148 ATG proteins and autophagy regulators that have the potential to affect carcinogenesis through modulating autophagy selectivity. To the best of our knowledge, the identification of STBD1 W203C, ATG4B Y8C, and many others in our database

**Fig. 6 Depletion of STBD1 promotes glycolysis in colorectal cancer cells. a** The number of metabolites identified in each sample by targeted metabolomic profiling. Three biological replicates were performed. **b** Heatmap showing metabolites in several major pathways detected by targeted metabolomic profiling, from shSTBD1 and shControl HCT116 cells, respectively. For each metabolite, its levels in the six samples were normalized using the $z$-score method. **c–f** shControl and shSTBD1 HCT116 cells were cultured with $^{13}C_6$-glucose-containing medium for 12 h, and then cells were harvested for analysis by LC–MS/MS. 3-PG 3-phosphoglycerate, 2-PG 2-phosphoglycerate, PEP phosphoenolpyruvate, α-KG α-Ketoglutarate. **g** Mapping metabolites and genes whose abundance changed significantly in shSTBD1 HCT116 cells vs. shControl HCT116 cells to a pathway map. Metabolites and genes ($p < 0.05$) were shown. Filled circles, $^{13}C$-labeled carbon atoms; open circles, unlabeled carbon atoms; blue, downregulated; red, upregulated. **h** shControl and shSTBD1 HCT116 cells were cultured for 48 h, and then the medium was collected to determine the concentration of glucose and lactate concentrations. The lactate or glucose concentration was normalized to the total protein concentration, and the relative concentration was further normalized to that of the shControl HCT116 cells. **i** shControl and shSTBD1 HCT116 cells were cultured in a low glucose medium for 72 h. The cell viability was then assessed using the MTT assay and normalized to that of cells grown in a high glucose medium. **j** shControl and shSTBD1 HCT116 cells were incubated in indicated concentrations of 2-DG for 48 h. The cell survival rate in each group was evaluated by the MTT assay, and normalized to that of the control group (0 mM). Experiments **h–j** were performed in triplicate. Statistical data are presented as mean ± SD. Statistical comparisons were performed using an unpaired $t$ test. ***$p < 0.001$, **$p < 0.01$, *$p < 0.05$, ns (not significant), $p > 0.05$.

represents the first example. The wild distribution of such mutations in cancer samples suggests that altering of autophagy selectivity represents a common mechanism for the pathogenesis of multiple cancers.

STBD1 is the cargo receptor for glycogen autophagy, which is responsible for transporting glycogen into the lysosome to produce non-phosphorylated glucose[26,27]. Glycophagy is the glycogen breakdown pathway alternative to gluconeogenesis[45,46]. Although the functions of glycophagy in neonatal development, in which the gluconeogenesis machinery is not fully established, have been long appreciated, its roles in tumorigenesis were unclear[47–49]. By demonstrating that STBD1 and GAA have potential tumor-suppressive functions, we identify the previously uncharacterized connection between glycophagy and tumorigenesis. Depletion of STBD1 or disruption of its association with LC3 leads to enhancement of glycolysis and likely the pentose phosphate pathway in cancer cells, and promotes cancer growth. Our data are consistent with the observation that the expression of STBD1 is significantly downregulated in a diverse array of human tumors, and that the expression level of STBD1 is associated with cancer patients' survival probability (Fig. 3e, f). The discovery that 2-DG, a glycolysis inhibitor, significantly inhibits the growth of STBD1 low-expressing cancer cells, indicates that targeting glycolysis could represent an effective personalized targeting strategy for the patients with STBD1 low-expressing tumors (Fig. 6h).

Why does the inhibition of glycophagy contribute to tumorigenesis? Glycogen is degraded via two major pathways: the cytosolic pathway that decomposes glycogen into glucose-1-phosphate and glucose, and glycophagy that decomposes glycogen into glucose. Therefore, it is expected that glycophagy inhibition could cause metabolic reprogramming. Indeed, we find that the depletion of STBD1 increases the expression of multiple key glycolytic enzymes, and enhances the TCA cycle and nucleotide biosynthesis. Suppression of STBD1—either via knockdown or expression of the mutant—also promotes the expression of multiple cancer hallmark genes, including c-Myc, NFKB1, and AKT1, although the exact underlying mechanisms remain to be determined. We show that STBD1 suppression promotes cancer cell proliferation, detected by the MTT assay and the proliferation marker Ki67. It should be noted that the MTT assay measures reducing power, in particular, NADH in mammalian cells. Therefore, cautions must be taken to use the MTT assay when the metabolic states of cells are altered. Other methods of detecting cell proliferation, such as the detection of Ki67 and/or BrdU labeling, should be used at the same time.

Although we focus on testing the role of STBD1-mediated autophagy in cancer in this study, our rich dataset opens up the discovery of other uncharted autophagy pathways that regulate the development of cancer. For instance, TBC1D15 is an established mitophagy regulator and is also demonstrated as an oncoprotein

by a separate study[50–52]. However, it is unknown whether TBC1D15 exerts its pro-cancer activity through mitophagy. The identification of a cancer mutation within the LIR motif in TBC1D15 indicates that the disruption of TBC1D15 function in mitophagy may contribute to cancer development. Similarly, we have identified two Type II mutations with the LIR motif of TP53INP2 (tumor protein p53-inducible nuclear protein 2), indicating that TP53INP2 likely functions through bridging autophagy and apoptosis[53]. Furthermore, many proteins in our list have not been explored for their functions in cancer development, and delineation of their mechanisms could open up new directions for the field. One of such examples is TBC1D25, which is involved in the fusion of autophagosomes with endosomes and lysosomes[54]. It contains an LIR motif, which has two deleterious mutations in cancer samples based on our prediction. However, the functions of TBC1D25 in cancer development have not been explored. In summary, our work discovers a new tumorigenesis mechanism through the misregulation of autophagy selectivity. We expect that our study will benefit the discovery of novel autophagy-related pathways in cancer, and open new avenues that selectively target autophagy sub-routines for cancer therapeutics.

## Methods

**Data collection and preparation.** First, we search PubMed with a number of keywords, such as "LIR", "AIM atg8", "Atg8-family interacting", "LC3-interacting", and "LIR-containing". The full texts of all retrieved papers are carefully curated to collect experimentally identified LIR motifs. Also, we integrate 27 and 36 reported LIR motifs from iLIR[19] and hfAIM[20], respectively. To ensure the data quality, putative LIR motifs maintained in the iLIR database[55] and iLIR@viral[56] are not included. Then, we map known LIR motifs to primary protein sequences downloaded from UniProt database[36] to pinpoint their exact positions (On October 17, 2019). After redundancy clearance, we obtain 127 known LIR motifs including 121 cLIR and 6 atypical LIR (aLIR) motifs in 105 unique LIRCPs (Supplementary Data 1).

In this study, we hypothesize that short flanking peptides around core LIR motifs might be essential for interacting with LC3/Atg8, and we define a LIR motif peptide LMP(7, 7) as a cLIR or aLIR tetrapeptide flanked by 7 residues upstream and 7 residues downstream, with a total length of 18 aa to balance the training time and accuracy. Before model training, we regard LMP(7, 7) entries derived from all known cLIR and aLIR motifs as positive data, and we take LMP(7, 7) items around other putative cLIR motifs in the same proteins as negative data. Then, we construct a high-quality benchmark data set, containing 127 positive motifs and 931 negative motifs from 105 LIRCPs. For each known LIRCP, its gene names, UniProt accession number, LIR motif positions, LMP(7,7) item, species information, and PubMed IDs (PMIDs) of original references are present (Supplementary Data 1).

**Performance measurements.** To evaluate the accuracy of pLIRm, we calculate six measurements, including accuracy (Ac), sensitivity (Sn), specificity (Sp), positive predictive value (PPV), negative predictive value (NPV) and Mathew correlation coefficient (MCC) as below

$$Ac = \frac{TP + TN}{TP + FP + TN + FN} \quad (1)$$

$$Sn = \frac{TP}{TP + FN} \quad (2)$$

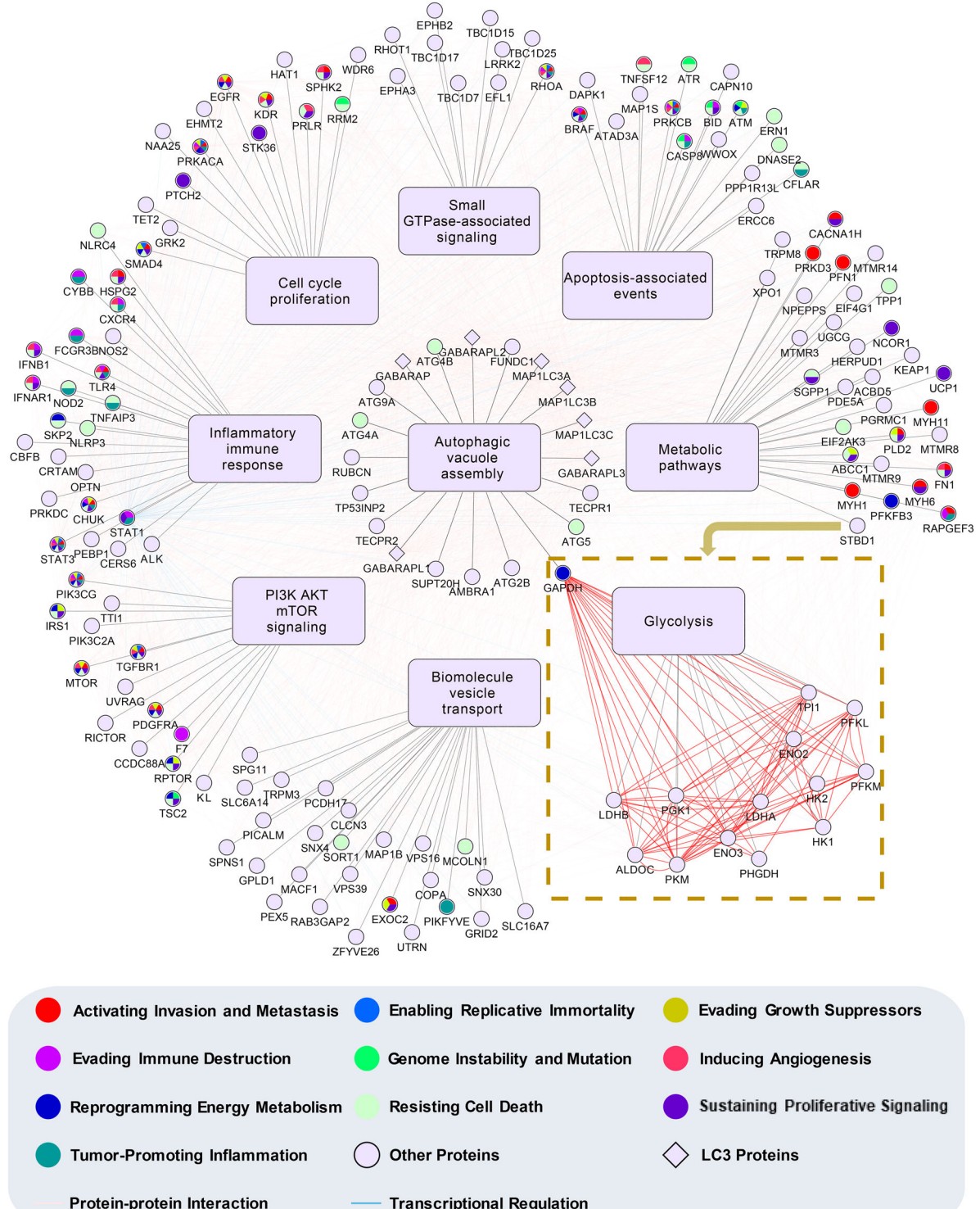

**Fig. 7 A LIRCP-regulating network that connects autophagy and carcinogenesis.** The 148 LAM-containing ATG proteins and autophagy regulators were classified into nine groups based on their major biological function. Both PPIs and transcriptional regulations were incorporated for these proteins if available. The downstream pathway, glycolysis, and corresponding proteins in the pathway regulated by STBD1 were also integrated.

$$\text{Sp} = \frac{\text{TN}}{\text{TN} + \text{FP}} \qquad (3)$$

$$\text{PPV} = \frac{\text{TP}}{\text{TP} + \text{FP}} \qquad (4)$$

$$\text{NPV} = \frac{\text{TN}}{\text{TN} + \text{FN}} \qquad (5)$$

$$\text{MCC} = \frac{(\text{TP} \times \text{TN}) - (\text{FN} \times \text{FP})}{\sqrt{(\text{TP} + \text{FN}) \times (\text{TN} + \text{FP}) \times (\text{TP} + \text{FP}) \times (\text{TN} + \text{FN})}} \qquad (6)$$

The LOO validation and 4-, 6-, 8-, and 10-fold cross-validations are conducted. The receiver operating characteristic curves are illustrated based on *Sn* and 1-Sp values, and the AUC scores are calculated.

**A modified GPS algorithm**. In 2004, we developed the GPS 1.0 algorithm for prediction of kinase kinase-specific phosphorylation sites[57]. Based on a hypothesis that similar short peptides might share similar biological properties and functions, we used an amino acid substitution matrix, e.g., BLOSUM62, to measure the sequence similarity among short peptides around known or putative phosphorylation sites[57]. We adopted this basic scoring strategy in all versions of GPS algorithms and incorporated more methods to improve the accuracy in later versions[58]. In GPS 5.0 (http://gps.biocuckoo.cn/)[17], we developed two additional approaches including PWD and SMO in order to improve the accuracy, besides the basic scoring strategy. PWD could efficiently optimize the position-specific weight values of short peptides around phosphorylation sites, whereas SMO could rapidly determine the scoring matrix. We used the PLR algorithm with the ridge (L2) penalty to optimize parameters[17].

Here, we modify the original GPS 5.0 algorithm to comprise two parts. In the part of the basic scoring strategy, we measure the average similarity score ($S$) of a given LMP(7, 7) item against all known LMP(7, 7) entries in positive data as below

$$S = \frac{1}{N} \sum_{j=1}^{L} (\sum_{i=1}^{N} M_f[P_j, K_{ij}]) \times W_j \tag{7}$$

Where $L$ is the length of the LMP(7, 7) item and equal to 18 in this study. $N$ is the number of known LMP(7, 7) items in positive data. $K_{ij}$ is the amino acid at position $j$ around a known LMP(7, 7) item $K_i$ ($i = 1, 2, 3, \ldots, N$). $W_j$ is the weight score of position $j$, and $M_f$ denotes the finally determined amino acid substitution matrix. In pLIRm, we only consider LMP(7, 7) items following the cLIR motif.

The performance improvement part comprises two steps, including PWD and SMO. The former is established based on the hypothesis that different positions in LMP(7, 7) items might differentially contribute to the recognition of LC3/Atg8, whereas the latter automatically generates an optimal amino acid substitution matrix to measure the sequence similarity of different LMP(7, 7) items. We adopt a refined PLR algorithm with two additional steps, including random mutation and random zeroing to determine parameters.

(i) PWD: We initially use the amino acid substitution matrix BLOSUM62 ($M_{BLOSUM62}$) to calculate an average similarity score at the position $j$ of an LMP(7, 7) item $P$ as $S'_j$:

$$S'_j = W_j \frac{1}{N} \sum_{i=1}^{N} M_{BLOSUM62}[P_j, K_{ij}] \tag{8}$$

First, the weight score of each position $W_j$ in the LMP(7, 7) item is set to 1. Then the LOO validation is conducted to calculate the initial AUC value. The final $W_j$ vectors are computationally optimized by the refined PLR algorithm, based on the highest AUC value

$$W_j = W_1, W_2, W_3, \ldots, W_{18} \tag{9}$$

(ii) SMO: The average similarity value of an amino acid $a$ in the given LMP(7, 7) item $P$ and a residue $b$ in all known LMP(7, 7) entries in positive data is defined as $S_{ab}$:

$$S_{ab} = \frac{1}{N} \sum_{j=1}^{L} Q_j \times M_{BLOSUM62}[a, b] \times W_j \tag{10}$$

Where $Q_j$ is the number of $ab$ amino acid pairs at position $j$. In BLOSUM62, there are 24 types of characters, including 20 types of amino acids and 4 non-canonical characters (B, Z, X, and *). Thus, a number of ($[24 * (24 + 1)]/2 = 300$) unique $S_{ab}$ scores ($S_{ab} = S_{ba}$) are produced. Then, the updated PLR algorithm is used to optimize all of the $S_{ab}$ scores to generate the final matrix $M_f$

$$M_f = (S_{AA}, S_{AC}, S_{AD}, \cdots, S_{**})_{300} \tag{11}$$

(iii) The refined PLR algorithm. To optimize the parameters in PWD and SMO, the least absolute shrinkage and selection operator (LASSO, L1 regularization) penalty is used. In the step of random mutation, a parameter is randomly selected with +1 or −1 per time, and the result is adopted if the LOO validation AUC value is increased. To avoid the local optimization, the random zeroing step is added by randomly zeroing one parameter per time, and the manipulation is adopted if the AUC value is increased. The two steps are iteratively performed until the AUC value is not enhanced any longer. The improved PLR algorithm is implemented under Python 3.6 with Scikit-learn 0.21 (https://scikit-learn.org/stable/), a widely used machine-learning toolbox.

**Collection of human cancer mutations**. First, the human proteome set is downloaded from UniProt database[36], which contains 20,659 unique protein sequences. Then, we download human cancer mutations from the TCGA data portal (https://portal.gdc.cancer.gov/, level 4 data, in May 2018)[22]. All available projects are downloaded, and gene names in TCGA files are used to map the mutation data to human proteins. It should be noted that the clinical information is absent for the level 4 mutations. Also, we download all simple somatic mutations of ICGC release 28 from ICGC data portal (https://dcc.icgc.org/releases/release_28/Projects, in November 2019)[23]. For each project, the file "simple_somatic_mutation.open.*.tsv.gz" is downloaded. Ensembl gene IDs (columns entitled 'gene_affected' in ICGC data) are used to map ICGC mutations to human proteins. In addition, we obtain cancer mutations of COSMIC in file 'CosmicMutantExportCensus.tsv.gz' downloaded from the COSMIC website (https://cancer.sanger.ac.uk/cosmic/download, in Jan 2019)[24]. Ensembl transcript IDs are used to map COSMIC mutations to human proteins. In

total, we obtain 1,898,302, 4,306,716, and 5,806,067 human missense SNVs from TCGA, ICGC, and COSMIC databases, respectively. We merge the three data sets together, and there are 2,963,952 unique missense SNVs reserved after redundancy clearance.

**The collection of human ATG proteins and autophagy regulators**. In a previous study, we constructed a database named THANATOS (http://thanatos.biocuckoo.org/) in order to collect, curate, annotate, and maintain important proteins and post-translational modification events involved in regulating autophagy and cell death pathways[59,60]. Here, 43 known ATG proteins and 875 autophagy regulators in *Homo sapiens* are directly taken from THANATOS[59,60]. To avoid any missing autophagy regulators, we also obtain 58 well-known autophagy regulators from a previously published review[61]. After redundancy clearance, 911 human autophagy regulators are reserved.

**The pLAM algorithm**. From the human proteome set, we detect 19,577 proteins containing at least one tetrapeptide that follows the cLIR motif. Then, we define a LAM as a missense SNV located within an LMP(7,7) region that potentially influences the cLIR motif, and obtained 842,789 potential LAMs in 238,840 LMP(7,7) items of 18,806 potential LIRCPs. Then, we use the positive LMP(7,7) items $P$ and negative LMP(7,7) items $N$ to estimate the Bayesian posterior probability (BPP). For a given LMP(7,7) peptide $P_i$, the score $S_i$ calculated by pLIRm is transformed into a BPP value as below

$$p(P, |, S_i) = \frac{f(S_i, |, P)p(P)}{f(S_i, |, P)p(P) + f(S_i, |, N)p(N)} \tag{12}$$

As previously described[18], the prior probability scores of $p(P)$ and $p(N)$ reflect our belief in the distribution of $P$ and $N$ and are determined as the corresponding AUC value and 1, respectively.

Then, pLIRm is used to calculate the values for all potential LAMs ($n = 842,789$) before ($x_i$, $i = 1, 2, 3, \ldots, n$) and after ($y_i$, $i = 1, 2, 3, \ldots, n$) the mutation, whereas all scores are normalized into BPP values. To estimate the global probability density distribution of $x$ and $y$, we hypothesize that the joint probabilities within a very small interval might follow a Gaussian distribution. We use the Parzen window method[18], a nonparametric density-estimation approach, to conjugate the Gaussian distributions in all continuous small intervals to estimate the global distribution as below

$$f(x, y) = \frac{1}{nh^2} \sum_{i=1}^{n} \frac{1}{2\pi} \exp\left[ -\frac{(x - x_i)^2 + (y - y_i)^2}{2h^2} \right] \tag{13}$$

Where $h$ is the window width and the size of $h$ influences the accuracy of the probability density estimation. The maximum likelihood estimation (MLE) method is used to determine the optimal $h$ value as below

$$f\left[ (x_1, y_1), (x_2, y_2), \cdots, (x_n, y_n) | h \right] = f\left[ (x_1, y_1) | h \right] \times f\left[ (x_2, y_2) | h \right] \times \cdots \times f\left[ (x_n, y_n) | h \right] \tag{14}$$

For different $h$ values (from 0 to 1, 0.0001 per step), we estimate the joint probability density distributions for each LAM from the remaining ones, until all LAMs are used once. The likelihood value is the product of $n$ probability density values, and the optimal $h$ value that maximized the likelihood value is determined as 0.0018 in this study.

For a given $x$ score, the probability density distribution of its corresponding $y$ values is estimated, and the statistical significance of each $y$ value is calculated, respectively. Under a threshold of $p$ value < 0.01, there are 74,319 potential LAMs that significantly change 50,671 cLIR motifs in 14,717 proteins. In order to distinguish the different impacts of LAMs on changing cLIR motifs, the mutated score $y$ is chosen as >0.5 for LAMs that potentially increase the binding affinity of cLIR motifs to LC3 (Type I, $y > x$), and the original score $x$ is selected as >0.5 for LAMs that potentially decrease the binding affinity (Type II, $x > y$). Then, 935 Type I and 2595 Type II LAMs that significantly influence 2942 cLIR motifs in 2561 proteins are reserved, respectively. Finally, we map these LAMs to known ATG proteins and autophagy regulators so as to prioritize the 148 LAM-containing LIRCPs potentially involved in human cancer through regulating autophagy.

**The enrichment analyses**. The hypergeometric test is adopted for the enrichment analysis of 148 predicted LIRCPs (Supplementary Data 3). Here, we define

$N$ = number of human proteins annotated by at least one term
$n$ = number of human proteins annotated by term $t$
$M$ = number of LIRCPs annotated by at least one term
$m$ = number of LIRCPs annotated by term $t$

Then, the enrichment ratio (E-ratio) is calculated, and the p value is computed with the hypergeometric distribution as below

$$E - \text{ratio} = \frac{\frac{m}{M}}{\frac{n}{N}} \quad (15)$$

$$p\,\text{value} = \sum_{m'=m}^{n} \frac{\binom{M}{m'}\binom{N-M}{n-m'}}{\binom{N}{n}}, (E - \text{ratio} > 1) \quad (16)$$

In this study, only statistically enriched GO terms and KEGG pathways are considered. GO annotation files (on 10 October 2019) have been downloaded from the Gene Ontology Consortium Web site (http://www.geneontology.org/), and we obtain 19,714 human proteins annotated with at least one GO biological process term[62]. KEGG annotation files (released on 4 December 2017) have been downloaded from the ftp server of KEGG (ftp://ftp.bioinformatics.jp/)[63], which contains 6956 human annotated genes.

**The TCGA data with clinical outcomes**. We download human cancer mutations (*Mutation_Packager_Calls.Level_3.*), mRNA expression levels (*.mRNAseq_Preprocess.Level_3.*), DNA methylation profiles (*.Merge_methylation_*.Level_3.*), and clinical outcomes (*.Merge_Clinical.Level_1.*) from BROAD Institute (http://gdac.broadinstitute.org/runs/stddata__latest/)[22]. All projects on 37 types of cancer are downloaded, including adrenocortical carcinoma, bladder urothelial carcinoma, breast invasive carcinoma, cervical and endocervical cancers, cholangiocarcinoma, colorectal adenocarcinoma, colon adenocarcinoma, lymphoid neoplasm diffuse large B-cell lymphoma, oesophageal carcinoma, GBMLGG, glioblastoma multiforme (GBM), head and neck squamous cell carcinoma, kidney chromophobe (KICH), pan-kidney cohort (KICH+KIRC+KIRP) (KIPAN), kidney renal clear cell carcinoma (KIRC), kidney renal papillary cell carcinoma (KIRP), acute myeloid leukaemia, LGG, liver hepatocellular carcinoma, lung adenocarcinoma, lung squamous cell carcinoma, mesothelioma, ovarian serous cystadenocarcinoma, pancreatic adenocarcinoma, pheochromocytoma and paraganglioma, prostate adenocarcinoma, rectum adenocarcinoma, sarcoma, skin cutaneous melanoma, stomach adenocarcinoma, stomach and esophageal carcinoma, testicular germ cell tumours, thyroid carcinoma, thymoma, uterine corpus endometrial carcinoma, uterine carcinosarcoma, and uveal melanoma. In total, we obtain 798,478 missense SNVs with clinical outcomes in 11,148 tumor samples across 37 major cancer types/subtypes. The RNA-Seq by Expectation-Maximization (RSEM) values of genes are reserved, as well as beta values of methylation data. Gene names are used to map the TCGA data to human LIRCPs.

**Survival analyses**. First, we analyze the association between missense SNVs and clinical outcomes of each cancer type. For each LAM-containing ATG protein and autophagy regulator, patients with or without at least one cancer-derived SNV are classified into two groups and compared with the two-sided log-rank test (p value < 0.05). The Kaplan-Meier survival curves are illustrated by the R package survminer v0.4.6 with the function of "ggsurvplot" (http://www.sthda.com/english/rpkgs/survminer/). For each gene in the RNA-seq data, we adopt the Cox proportional hazard model for overall survival data, and we stratify patients of each cancer type into two groups as high and low mRNA expression. The two-sided log-rank test is performed (p value < 0.05), and the Kaplan–Meier survival curves are plotted by ggsurvplot. The same procedure is also conducted for analyzing DNA methylation data (p value < 0.05).

**Cell culture and transfection**. HEK293T, H1299, A549, HCT116, HeLa, A2780, MDA-MB-468, MGC803, HGC27, HT29, LO2, HepG2, MDA-MB-221, and H460 cells are obtained from the American Type Culture Collection (ATCC) and Cell Bank of Chinese Academic of Sciences (Shanghai, China), respectively, and cultured in Dulbecco's Modified Eagle Medium (DMEM, Gibco), supplemented with 10% fetal bovine serum (FBS, Excellbio) and penicillin–streptomycin (Hyclone) at 37 °C in 5% CO2. For exogenous expression, cells are transfected with polyethylenimine (PEI, Sigma) in serum and antibiotic-free medium, according to the manufacturer's instruction. The medium is changed after 4–6 h, and cells are harvested after 48 h from transfection.

**DNA constructs and mutagenesis**. PCR-amplified STBD1 is cloned into pcDNA3.1(+), mCherry-N1, or plvx-ires-neo. PCR-amplified ATG4B is cloned into pcDNA3.1(+). Mutations are generated using the High-Fidelity PCR kit (MACLAB).

**Co-immunoprecipitation**. Cells are harvested and lysed in lysis buffer (50 mM Tris, pH 7.4, 50 mM NaCl, 0.5% Nonidet P-40) containing protease inhibitors (Bimake), similar to previous studies[64,65]. Lysates are centrifuged at 14,000g for 20 min at 4 °C. Anti-FLAG gel (Bimake) is equilibrated with tris buffered saline (TBS) before use. The supernatant from cell lysates is mixed with affinity gel and incubated at 4 °C overnight. The samples are centrifuged at 5000 g for 30 s to pellet the sepharose. The sepharose gel is washed by TBS three times, and sodium dodecyl sulfate loading buffer is added for sodium dodecyl sulfate–polyacrylamide gel electrophoresis and immunoblotting.

**Tissue microarray, IHC, and staining intensity analysis**. A tissue microarray chip containing 28 pairs of human colon cancer tissues and matched adjacent normal colon tissues is purchased from Shanghai Outdo Biotech CO., LTD. (Shanghai, China, HColAde060CS01). The expression of STBD1 is evaluated with the anti-STBD1 antibody (proteintech, 67018-1-Ig) by IHC at a dilution of 1:200. The staining substrate was 3,3′-diaminobenzidine (DAB) and the nuclei is counterstained with hematoxylin. Then the tissue microarray chip is scanned using Pannoramic MIDI (3D HISTECH). The STBD1 expression was performed based on staining intensity by analyzing the IOD of the dark brown color in each image using NIH Image J software.

**Fluorescence microscopy**. Immunofluorescence experiments are performed as previously reported[66,67]. HeLa cells are seeded onto circular coverslips in 24-well plates and then transfected with indicated plasmids. After 24 h, cells are fixed with 4% paraformaldehyde and permeabilized with 0.1% Triton X-100, blocked using 5% FBS in PBS, and then incubated with anti-glycogen monoclonal antibody (IV58B6) at 4 °C overnight[68,69]. Cells are then stained with Alexa Fluor 647 goat anti-mouse IgG for 8 h at 4 °C and then incubated with Hoechst 33258 (Sigma). After washing the cells three times in PBS, the images are captured using the Olympus FV-1000 confocal microscopes and analyzed using NIH Image J software.

**Glycogen assay**. The glycogen levels are measured using the glycogen assay kit (Solarbio), according to the manufacturer's instructions. Briefly, $5 \times 10^6$ cells are collected to ultrasonication and heated for 20 min at 95 °C. After 10 min of centrifugation at 8000g, the supernatants are used to measure the level of glycogen. The glycogen concentration is normalized to the cell protein concentration.

**Lentiviral production and infection**. HEK293T cells are co-transfected with lentiviral vectors expressing non-target control shRNA, shSTBD1, STBD1 WT, or STBD1 W203C along with pHCMVG, pMDLg/PRE, and pRsv-Rev (shRNA/plvx: HCMVG:pMDLg/PRE:pRsv-Rev = 3:1:1:1). After 48 h, the viral supernatants are collected, filtered, and added with 4 µg/ml polybrene (YEASEN) to infect target cells. After 72 h of infection, the target cells are cultured in a selection medium containing 2 µg/ml puromycin (BBI life sciences) or 500 ng/µl G418 (Diamond). Afterward, a fraction of cells is harvested to determine the knockdown efficiency or overexpression of STBD1 using immunoblotting.

For CRISPR–Cas9-mediated gene knockout, gRNA sequences are introduced into V2T construct via PCR. Constructs encoding Cas9 and gRNA are cotransfected with viral packaging plasmids (V2T:psPAX2:pMD2.G = 2:1:1) into HEK293T cells. After 48 h, the viral supernatants are collected, filtered, and added with 4 µg/ml polybrene (YEASEN) to infect A549 cells. After 72 h of infection, the target cells are cultured in a selection medium containing 2 µg/ml puromycin (BBI life sciences). Afterward, a fraction of cells are harvested to determine the KO clones using immunoblotting. The sequences of the short hairpin RNAs and gRNAs are shown as below

shControl: TTCTCCGAACGTGTCACGT
shSTBD1: GCAATGGACATTTGATTTCTA
shATG4B: CAGCGTCCTCAACGCATTCAT
STBD1 gRNA 1: TAAAGTGGTTCACGCATGGT
STBD1 gRNA 2: GAATGGGGGAGTTACCCGCT

**Cell proliferation assay**. For cell proliferation assay, cells are seeded in 96-well culture plates at 1000 per well. After 72 h (or 72 h after transfection with siRNAs), cells are evaluated by the MTT assay kit (Servicebio). The absorbance is measured at OD 570 nm. For colony formation, 1000–2000 stably expressing indicated genes cells are seeded in 6-well culture plates. After 20 days of culture, the colonies are fixed with fixing solution for 15 min and stained with 0.5% crystal violet for 15 min, then washed with ddH2O. For colony formation after GAA siRNAs treatment, 500–1000 cells are seeded in 12-well culture plates and transfected with 3 different siRNA molecules (siControl, siGAA-1, or siGAA-2), respectively every 3 days. After 12 days of culture, the colonies are fixed with fixing solution for 15 min and stained with 0.5% crystal violet for 15 min, then washed with ddH2O. The number of colonies is analyzed by NIH Image J software.

**Tumor xenograft**. Human tumor xenografts are established using an established protocol[70]. Briefly, fresh HCT116 cells ($10^7$ per mouse, re-suspended in 200 µl of PBS) are subcutaneously injected into the flank of 5-week-old BALB/c nude mice (Charles River). The tumor volumes are measured every 2 days using calipers and calculated using the equation (length × width$^2$)/2. 15 days after injection, the mice are euthanized and tumors are isolated and weighed. Tumors are divided into two factions, one for analysis by IHC and one for analysis by immunoblotting. For IHC, the formalin-fixed, paraffin-embedded tumor sections are stained with antibodies against c-Myc and Ki67 (Servicebio, GB13030-2). The tumor apoptosis is measured by TUNEL staining using a TUNEL assay kit (Servicebio), followed by DAPI counterstaining. The images are captured using the Olympus microscope, and the integrated density or the percentage of cells staining positively for TUNEL is determined using NIH Image J software. For immunoblotting, a fraction of tumors is extracted in RIPA buffer (Beyotime), which then detected the expression of c-Myc, AKT1 (CST, 2938), NF-κB1 (CST, 3035), STBD1 (proteintech, 10828-1-AP),

and Tubulin (proteintech, 11224-1-AP). The animal welfare and experiments conform to the guidelines for care and use of laboratory animals and are performed according to the guidelines and approval of the Animal Investigation Committee of the West China Second University Hospital, Sichuan University.

**siRNA knockdown assay**. Cells are seeded at 12-well culture plates and transfected with siRNAs and non-targeting control siRNA using Lipofectamine RNAiMax Reagent (Invitrogen) in OptiMEM medium, according to the manufacturer's instruction. The medium is changed after 4-6 h, and cells are harvested 72 h after transfection. Knockdown efficiency is measured by qRT-PCR.

siControl: UUCUCCGAACGUGUCACGUTT
siGAA-1: CCUCCACUUCACGAUCAAATT
siGAA-2: GGAAUAACACGAUCGUGAATT

**Gene expression profiling of cancer cells**. Totally, $5 \times 10^6$ HCT116 cells are harvested and lysed by TRIzol (Invitrogen). Total RNA is extracted according to the manufacture's protocol. RNA is then analyzed for purity by measuring the ratio of absorbance at 260 and 280 nm using the NanoPhotometer Spectrometer (IMPLEN CA, USA). The cDNA libraries are synthesized using NEBNext UltraTM RNA Library Prep Kit for Illumina (NEB, USA), according to the manufacture's protocol. The size of the library insert fragments is measured by Agilent 2100 bioanalyzer. The suitable library fragments are sequenced on an Illumina platform using 125 bp/150 bp paired-end technology.

To obtain clean data with high quality for downstream analysis, clean data are obtained by removing reads containing adapter, poly-N and low-quality reads from raw data. Reference genome (Hg38) and gene model annotation files (release 93) are downloaded from ENSEMBL. Clean reads are aligned to the reference genome using STAR 2.7.1a. The reads counts mapped to each gene are counted using R package GenomicAlignments v1.18.1. Gene expression abundance are calculated by RSEM v1.3.1, and transcripts per kilobase of exon model per million mapped reads (TPM) of each gene is calculated based on the length of the gene and reads count, and used to represent gene expression abundance. Differential expression analysis of two groups (three biological replicates) is performed using DESeq2 1.22.2. Genes with an | TPM fold change | >1 and $p$ value < $10^{-5}$ are set as differentially expressed. KEGG pathways enrichment analysis is performed by R package clusterProfiler v3.10.1, and pathways with $p$ value < $10^{-4}$ are considered as significantly enriched as DEGs.

For the enrichment analysis of cancer hallmarks affected by STBD1 depletion, 1611 DEGs with a relaxed stringency are adopted (| TPM fold change | >1 and $p$ value < 0.01). From a well-curated resource named HOC database[30], we obtain known cancer hallmark genes that cover 11 hallmarks of cancer, including evasion of anti-growth signaling, replicative immortality, sustained proliferation, genome instability, cell death, immune evasion, invasion and metastasis, microenvironment, angiogenesis, metabolism, and inflammation. Because multiple types of cells beyond cancer cells are involved in forming the tumor microenvironment[71], this hallmark is not considered. Then, we map the 377 genes in the remaining 10 hallmarks to the transcriptomic data, and the hypergeometric test is performed for each hallmark using the 1611 DEGs ($E$-ratio > 1, $p$ value < 0.01).

Differential expression analysis of shSTBD1/shControl, W203C/WT and WT/plvx neo (three biological replicates) is performed using DESeq2 1.30.0. Genes with relaxed $p$ value < 0.05 are set as differentially expressed. KEGG pathways enrichment analysis is performed by using a hypergeometric test, and pathways with $p$ value < 0.005 are considered as significantly enriched by DEGs.

**Real-time PCR**. Totally, $3 \times 10^6$ shControl HCT116 cells and shSTBD1 HCT116 cells are harvested and lysed by TRIzol (Invitrogen). Total RNA is extracted according to the manufacture's protocol. RNA is then analyzed for purity by measuring the ratio of absorbance at 260 and 280 nm using the NanoPhotometer Spectrometer (IMPLEN CA, USA). cDNA libraries are synthesized using One-Step gDNA Removal and cDNA Synthesis SuperMix (TransScript, China) according to the manufacture's protocol. RT-PCR assay is carried out using 2× SYBR Green qPCR Master Mix (Bimake, China), and the threshold cycle (Ct) value is measured. TUBB is used as the housekeeping gene for normalizing genes, respectively. The comparative gene expression is calculated with the $2^{-\Delta\Delta CT}$ method. All primers used are synthesized from Sangon Biotech (Shanghai, Supplementary Data 9).

**Metabolomic profiling of cancer cells**. HCT116 cells stably expressing shControl or shSTBD1 established as described above are cultured in DMEM supplemented with 10% FBS for 24 h. Then shControl and shSTBD1 HCT116 cells ($1 \times 10^7$) are harvested and extracted using pre-chilled 80% ($v/v$) methanol as previously done[72]. Briefly, the culture medium is removed, and cells are washed twice with ice-cold PBS and extracted using pre-chilled 80% methanol. The lysates are then concentrated at 14,000$g$ for 15 min, and the supernatant is collected to dryness using a SpeedVac (Labconco, USA). The metabolites are re-dissolved using 80% methanol and an injection volume of 3 μL is used for LC-MS/MS analysis. Metabolites are analyzed by TSQ Quantiva (Thermo, CA) with an Ultimate 3000 LC system (Thermo Fisher Scientific, USA). Data analysis and quantitation are performed by the software TraceFinder 3.2 (Thermo Fisher, CA). For plotting the heatmap, the

original expression levels of each metabolite in the 6 samples are normalized using the $z$-score transformation, one of the most used normalization methods[73]. The mean expression value $\mu$ and standard deviation (SD) $\delta$ are calculated. For the metabolite $i$ with the expression level of $x_i$, its normalized $z$-score is calculated as below

$$z_i = \frac{x_i - \mu}{\delta} \qquad (17)$$

For $^{13}C_6$-labeled metabolites analysis, shControl and shSTBD1 HCT116 cells are cultured in glucose-free DMEM (Gibco) supplemented with 2.25 g/L $^{13}C_6$-glucose and 2.25 g/L unlabeled-glucose for 12 h. The cells ($1 \times 10^7$) are then harvested and extracted using pre-chilled 80% ($v/v$) methanol as above. Unlabeled cells are cultured in parallel in DMEM (Gibco) with equal concentrations of unlabeled glucose to identify unlabeled metabolites. The metabolites are analyzed by TSQ Quantiva (Thermo, CA) coupled with Ultimate 3000 (Thermo Fisher Scientific, USA). MS/MS spectra are acquired with stepped NCE of 15, 30, and 45. Data analysis and quantitation are performed by the software TraceFinder 3.2 (Thermo Fisher, CA).

**Lactate secretion and glucose consumption**. Cells are seeded in 24-well culture plates at 5000 per well, after 48 h cell mediums are collected for assay. Lactate secretions are measured using a lactate assay kit purchased from Nanjing Jiancheng Bioengineering Institute and glucose levels are assessed by glucose assay kit (Robio) according to the manufacturer's instructions. The lactate or glucose concentration is first normalized to protein concentration, and the relative concentration is further normalized to that of shControl HCT116 cells.

**Detection of cell survival rate**. After Cells are seeded in 96-well culture plates at 4000 per well overnight, 2-DG (2-deoxy-d-glucose, Selleck) in different concentrations (0, 5.7, 16.7, and 50 mM) is added for incubation at 37 °C and 5% $CO_2$ for 48 h. The cell survival rate in each group is evaluated by the MTT assay kit and normalized to that of the control group (0 mM).

**Re-construction of the LIRCP-regulating Network**. From the 8 public databases including ARN[32], BioGrid[37], IID[38], inBio MapTM[39], Mentha[40], HINT[41], iRefIndex[42], and PINA[43], we collect 1,771,193 PPIs and 131,541 transcriptional regulations of 18,839 human proteins from these databases, and map them to the 148 identified LIRCPs, 7 LC3 proteins and 14 proteins regulated by STBD1. Among the 91 transcriptional regulations, there are 88 for STAT1 and 3 for STAT3, respectively. Both two transcription factors, STAT1 and STAT3, are important autophagy regulators[73,74]. Similar to previous studies[33,34], the LIRCP-regulating network is constructed and visualized with Cytoscape 3.7.2 software package[75].

**Statistics and reproducibility**. All experiments are performed at least in triplicate. No statistical method is used to predetermine sample size and no data are excluded from the analyses. All statistical data are presented as the mean ± SD. Statistical significance of the difference is determined using Student's $t$ test. Differences are considered significant at the $p$ value < 0.05.

**Reporting summary**. Further information on research design is available in the Nature Research Reporting Summary linked to this article.

## Data availability

The RNA-seq data is deposited into GEO with the accession code GSE173273 and GSE173274 (secure: qjqdqioirdwptsl; secure: yfwtkymyrxyzjwx). The human cancer mutations data, mRNA expression levels, DNA methylation profiles, and clinical outcomes referenced during the study are available in public repositories from the TCGA data portal (https://portal.gdc.cancer.gov/, level 4 data, in May 2018), ICGC website (https://dcc.icgc.org/releases/release_28/Projects, in November 2019), COSMIC website (https://cancer.sanger.ac.uk/cosmic/download, in Jan 2019) and BROAD Institute (http://gdac.broadinstitute.org/runs/stddata__latest/). The source data underlying Figs. 3,4,5,6 and Supplementary Figs. 3–10 are provided as a Source Data file. All the other data supporting the findings of this study are available within the article and its supplementary information files and from the corresponding author upon reasonable request. Source data are provided with this paper.

## Code availability

The source codes of pLIRm and pLAM are submitted to GitHub (https://github.com/BioCUCKOO/pLIRm-pLAM). The data sets including 105 LIRCPs with 127 known LIR motifs and 18,806 human proteins containing at least one LAM are also submitted to GitHub for ensuring the reproducibility of the analyses.

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

## Acknowledgements

This research is supported by Special Project on Precision Medicine under the National Key R&D Program (2018YFC1005004, 2017YFC0906600 and 2018YFC0910500), Natural Science Foundation of China (91854121, 31871429, 31930021, 31970633, and 31671360), Sichuan Science and Technology Program (2018RZ0128), Fundamental Research Funds for the Central Universities (2017KFXKJC001, 2019kfyRCPY043, and 2020JYCXJJ027), Changjiang Scholars Program of China, and the program for HUST Academic Frontier Youth Team.

## Author contributions

Z.H. performed all the cellular and animal studies with assistance from Z.S. and X.S. and created Fig. 1, W.Z. carried out computational work with assistance from W.N., C.W., and W.D. and created Fig. 5f, and Z.L. performed RNA-seq analysis under supervision of L.C. X.L. helped the metabolic analysis, and O.B. and T.M. provided the glycogen antibody. Y.X. and D.J. conceived and supervised the project. Z.H., W.Z., Y.X., and D.J. prepared the paper with input from all authors.

## Competing interests

The authors declare no competing interests.
