## [Peer Review File · Nature Communications]

REVIEWER COMMENTS

Reviewer #1 (Remarks to the Author); expert on cancer metabolism and autophagy:

The authors have generated a pipeline for uncovering mutations in LIR motifs across the proteome that may have functional significance in modulating autophagy in cancer cells. Through this analysis several putative proteins containing mutated LIRs were identified and validated for binding to ATG8 homologs including STBD1 – a known autophagy receptor for glycogen. The authors find that the LIR in STBD1 (WEMV located at aa 203) is mutated to CEMV leading to loss of binding to GABARAPL1. Differential levels of STBD1 was also found in diverse cancer cell lines. Knockdown of STBD1 led to changes in glycogen co-localization with GABARAPL1, cellular metabolism, and in vitro and vivo tumor growth. The authors propose that STBD1 is a novel tumor suppressor.

Overall the computation pipeline appears to be an exciting new approach to identify novel LIR motifs and those that are altered in cancer. The data associated with BRAF for instance, showing a gain of binding to LC3 and GABARAPL1 is very interesting. However, a clear connection between the selected LIR mutation in STBD1 and autophagy regulation or tumorigenesis is not well established. Specific points are noted below.

Western blots for LC3 shown in extended figure 1 are conducted using overexpression of WT or mutant STBD1. The results as presented do not establish whether STBD1 loss changes autophagy flux, which should be conducted using established reporters or treatment with lysosome inhibitors.

In figures 4a and extended data figure 1b expression of STBD1 W203C causes a dramatic change in GABARAPL1 cellular distribution. Can the authors provide an explanation for this result and whether this phenotype contributes to the downstream effects observed ?

Based on the data presented in supplementary table 3 it appears that the W203C mutation is present in a single tumor sample (intestinal adenocarcinoma). It is difficult to therefore assign this gene as a bone fide tumor suppressor given the absence of wide spread inactivating mutations either in this cancer type or across cancer types. Therefore, the authors may wish to reconsidering assigning STBD1 as a tumor suppressor.

The patient survival data presented in fig. 3e, while statistically significant, does not convincingly show a major difference in the survival outcomes of patients with high vs low mRNA levels of STBD1. Is it possible that assessing survival within a given cancer type might show more convincing differences?

In order to establish cancer relevance, I believe it is important to validate the reported

reduced expression of STBD1 in cell lines (ext data figure 2c), in patient tumor specimens and provide statistical analysis of staining intensity. Moreover, it is important to include normal cell controls to assess relative change in STBD1 expression levels in cancer cells.

The authors interchangeably utilize the STBD1 W203C mutant and complete knockdown/knockout of STBD1 which are technically not equivalent. For instance the in vivo data presented in figure 4o,p are conducted using cells knocked down for STBD1. Does expression of STBD1 W203C mutant in STBD1 KO cells also accelerate tumor growth ?

The connection between STBD1 and Myc is difficult to reconcile. It is unclear as to why myc levels change in response to STBD1 knockdown or expression of the W203C mutant. Can the authors provide an explanation as to how and why this occurs ?

Related to the previous point, a direct role for STBD1 in regulation of metabolism or expression of metabolic genes is not convincingly established – the data as presented is largely correlative. For instance, is it possible that the increased glycolysis associated with STBD1 loss or expression of the W203C mutant may be indirect – ie associated with changes in Myc levels? If so additional mechanistic insight into how STBD1 changes Myc activity is important to establish.

Overall the study delves more so into the functional significance of STBD1 rather than a specific role for the LIR mutant and its effect on tumorigenesis, metabolism and autophagy.

Reviewer #2 (Remarks to the Author); expert on autophagy and computational biology:

The manuscript of Han et al describes a novel bioinformatics approach to identify autophagy targeted proteins, focuses on those that are mutated in cancer, and experimentally validates and assess one of the key predictions, STBD1. The manuscript is well written compared to the complexity of the study, though there are a few minor issues with the text listed below. The topic is very timely, and the applied approach extends our current knowledge. The assessment of the data has been carried out properly, except for a few points in relation for example to how the cancer-associated mutations were defined and used (see details below).

Major points:

- Title does not represent fully the study. It should be more specific to the presented work.
- The abstract does not contain the key results properly in relation to the STBD1 experiments. Ie, the paper contains much more than what is written currently in the abstract.

- Throughout the manuscript abbreviation usage is not ideal. There are much more abbreviations than needed, any many are not standard ones. For example “AA” as autophagy-associated (AA) proteins is not really common, and confusing with ATG proteins, even if AA meant to cover more. I suggest reducing the number of abbreviated terms.

- The mutation data, while described in detail in the Methods section, is not always clear in the Results section which one was used and why. For example in the Results section: “Next, we mapped known cancer mutations to potential human LIRCPs “. In the Methods there are three mutation sources, which one was used here, and why? Also, a more important point currently not covered is what was the mutational rate for the identified LAMs? How frequent they are in the different cancer types? COSMIC contains for example so many passenger mutations that should not be considered in general as cancer-associated. Therefore, more data presentation and quality control are needed here, before applying their new model.

- At some parts, the data integration is not always justified or clear. For example, “The pan-cancer analysis revealed that SNVs, RNA expressions, and DNA methylation levels of 19, 101 and 109 LIRCPs are statistically associated with human cancer (Fig. 3a).” Why these data is relevant when LIR motif based associations would affect the protein level and not the regulation of the respected genes? If external data is aimed to use as in silico confirmation than proteomics difference for example from the Human Protein Atlas’s cancer dataset would be much more adequate.

- In general, the whole “LAM-containing LIRCPs play a potential role in human cancer “ section is out of scope or focus of this study. While relevance in cancer is an important thing to validate, these omics layer provide this just indirectly, not in relation to the LAMs.

- The section on EHMT2, ERCC6, and BRAF are inserted in the middle of the STBD1 analysis and validation. Probably it would be better before the STBD1. Also, from the current brief text it is not clear what the role of the newly discovered LIR motifs is in EHMT2, ERCC6, and BRAF? Especially the BRAF is very relevant to cancer biology. This should be presented better, or removed for the sake of the focus of the study.

- There is a network reconstruction work in the study, which is needed in principle but was not inserted, justified and carried out properly. This LIRCP-regulating network contains protein-protein interactions (PPIs) and transcriptional regulations among the 151 AA proteins. Why these interactions were added? This reconstruction study should be in the Results but currently it is referred only in the Discussion and the Methods. And additional issue is that the layout and visualization of the network on Figure 7 are not informative, unclear and not following best practices in the field (for example, please check the guidance here <https://doi.org/10.1371/journal.pcbi.1007244>)

- The discussion on the LIR prediction model is missing, more details on the benchmarking with other existing methods is needed, not only that the coverage of the new method is higher. How many LAMs can be found with the existing methods? How many putative LIR motifs (from iLIR database and iLIR@viral, which were correctly left out in the original data curation) can be found/confirmed with the new extended algorithm.

- The “A Modified GPS Algorithm “ section is difficult to understand. More explanation is needed to understand and assess this part.
- A potential key issue is that according to the Methods from the human proteome set, the authors detected 19,577 proteins containing at least one tetapeptide that follows the cLIR motif. As they checked 20,659 proteins, it would mean that the authors have found these motifs in nearly all human proteins which is worrying. Even if they apply a model to filter false positives, this initial step and data could question their approach and the results, thus this should be addressed.

Technical details:

- On page 25, add specific Uniprot release number and or date of download – year is not enough.
- On page 25, It is not clear what was the initial rationale to define the short flanking peptides and the LMP(7, 7).
- On page 25, It is also not clear how the positive and the negative datasets were compiled from this sentence: “we regarded LMP(7, 7) entries derived from all known cLIR and aLIR motifs as positive data, and we took LMP(7, 7) items around other putative cLIR motifs in the same proteins as negative data.”
- Where is the result of the “Performance Measurements” mentioned in the Methods (page 26)?
- On page 26, “Recently, we improved our previously-developed GPS algorithm from 2.1 to 5.0...” – it is not clear what 2.1 to 5.0 means.
- In the end of the same sentence on page 26 “in order to predict kinase-specific phosphorylation sites (<http://gps.biocuckoo.cn/>, published elsewhere).” – please add the exact reference.

Minor:

- In the abstract: “Further analysis confirms that LAMs in ATG4B, EHMT2 and BRAF that can alter interactions with LC3 and/or autophagic activities.” – remove “that”
- In the abstract: “Unexpectedly, STBD1, a poorly-characterized protein, inhibits tumor growth via modulating glycogen autophagy, while its cancer-linked mutation abolishes the cancer inhibitory function.” I suggest rewriting as the “unexpectedly” refers to the second part of the sentence.
- In the abstract: “... provides a fundamental framework to uncover the molecular landscape that drives carcinogenesis via modulating autophagy selectivity.” – I do not think it is a “molecular landscape”, rather a “molecular mechanism”.
- In the Introduction: “LIRCPs “ is not introduced or described what it means.
- “Our benchmark data set was much larger than iLIR (Kalvari et al., 2014) and hfAIM (Xie et al., 2016), which only collected 27 and 36 known LIR motifs, respectively “ – Probably add details or reference to Fig2a where the authors’ benchmark is quantified properly.
- “we identified 233 potential LAMs that significantly change 177 cLIR motifs in 151

LIRCPs, including 64 Type I and 169 Type II LAMs” – Please rewrite as “including” is not in a proper place of the sentence.

- “indicating that a strong correlation between autophagy and human cancer “ – remove “that”

- On page 17, “differentially expression genes (DEGs)” – change to “differentially expressed genes”

- On page 22, “Glycophagyis” -> Glycophagy is”

- On page 27., “tothe” -> “to the”

- On Figure 1: There is a typo in “tumor proliferation”

Tamas Korcsmaros

Detailed Responses to Reviewers' Comments

Reviewer #1:

1. The authors have generated a pipeline for uncovering mutations in LIR motifs across the proteome that may have functional significance in modulating autophagy in cancer cells. Through this analysis several putative proteins containing mutated LIRs were identified and validated for binding to ATG8 homologs including STBD1 – a known autophagy receptor for glycogen. The authors find that the LIR in STBD1 (WEMV located at aa 203) is mutated to CEMV leading to loss of binding to GABARAPL1. Differential levels of STBD1 was also found in diverse cancer cell lines. Knockdown of STBD1 led to changes in glycogen co-localization with GABARAPL1, cellular metabolism, and in vitro and vivo tumor growth. The authors propose that STBD1 is a novel tumor suppressor.

Overall the computation pipeline appears to be an exciting new approach to identify novel LIR motifs and those that are altered in cancer. The data associated with BRAF for instance, showing a gain of binding to LC3 and GABARAPL1 is very interesting. However, a clear connection between the selected LIR mutation in STBD1 and autophagy regulation or tumorigenesis is not well established. Specific points are noted below.

Ans: We thank the reviewers for the time and efforts spent evaluating our manuscript. In the past over two months, we have been working very hard performing experiments and revising the manuscript, to address these concerns.

2. Western blots for LC3 shown in extended figure 1 are conducted using overexpression of WT or mutant STBD1. The results as presented do not establish whether STBD1 loss changes autophagy flux, which should be conducted using established reporters or treatment with lysosome inhibitors.

Ans: Thanks a lot for the suggestions. We tested whether STBD1 affected degradation of p62 in the presence or absence lysosome inhibitor BafA1. Overexpression of STBD1 WT, but not W230C, induces degradation of p62, in the absence of BafA1 (new Supplementary Fig. 3f). The presence of BafA1, however, leads to the accumulation of p62 in both cells (new Supplementary Fig. 3f). These data indicate that cancer-associated mutation of STBD1 on W203 abrogates its binding to GABARAPL1 and impairs its functions in autophagy.

3. In figures 4a and extended data figure 1b expression of STBD1 W203C causes a dramatic change in GABARAPL1 cellular distribution. Can the authors provide an explanation for this result and whether this phenotype contributes to the downstream effects observed?

Ans: STBD1 has a glycogen-binding domain, in addition to its LIR motif. We and others have shown that STBD1 has strong co-localization with glycogen. Therefore, GABARAPL1 co-localizes with glycogen through its interaction with STBD1 WT. STBD1 W203C disrupts the interaction with GABARAPL1, thus altering its cellular distribution. Although alteration of both GABARAPL1 cellular distribution and cell proliferation are induced by STBD1 LIR mutation, we have no evidence that they are directly connected with each other.

4. Based on the data presented in supplementary table 3 it appears that the W203C mutation is present in a single tumor sample (intestinal adenocarcinoma). It is difficult to therefore assign this gene as a bone fide tumor suppressor given the absence of wide spread inactivating mutations either in this cancer type or across cancer types. Therefore, the authors may wish to reconsidering assigning STBD1 as a tumor suppressor.

Ans: Thanks for the suggestion. We are no longer assigning STBD1 as a tumor suppressor, as substantially more evidence is needed to claim a tumor suppressor (as the reviewer has suggested). Instead, we suggest that STBD1 has potential tumor-suppressive activities based on our results. In addition to the point mutation in a single tumor sample, we also provide new evidences that STBD1 is down-regulated in liver cancer tissues relative to the adjacent non-carcinoma tissues (new Supplementary Fig. 4a).

5. The patient survival data presented in fig. 3e, while statistically significant, does not convincingly show a major difference in the survival outcomes of patients with high vs low mRNA levels of STBD1. Is it possible that assessing survival within a given cancer type might show more convincing differences?

Ans: In the original form, actually the survival analysis was performed in both pan-cancer and individual cancer levels, and here we added a new Supplementary Data 4 to present all statistically significant results (The log-rank test, SNV: p -value < 0.05 ; RNA expression: p -value $< 10^{-4}$; DNA methylation: p -value $< 10^{-4}$). For STBD1, we found that its higher mRNA expression level in glioma (GBMLGG) and lower DNA methylation level in GBMLGG and brain lower grade glioma (LGG) are significantly associated with a lower survival probability, exhibiting an opposite result against that in the pan-cancer level. So in general STBD1 might have tumor suppressive functions, and it might have different roles in different types of cancer.

6. In order to establish cancer relevance, I believe it is important to validate the reported reduced expression of STBD1 in cell lines (ext data figure 2c), in patient tumor specimens and provide statistical analysis of staining intensity. Moreover, it is important to include normal cell controls to assess relative change in STBD1 expression levels in cancer cells.

Ans: Suggestions are well-taken. We performed immunohistochemistry (IHC) staining to examine 27 colon cancer specimens and paired adjacent noncancerous tissues. The expression of STBD1 at the protein level is significantly lower in the cancer tissues in comparison with that in the adjacent non-carcinoma tissues (~56% of paracancer) (new Supplementary Fig. 4a).

7. The authors interchangeably utilize the STBD1 W203C mutant and complete knockdown/knockout of STBD1 which are technically not equivalent. For instance the in vivo data presented in figure 4o,p are conducted using cells knocked down for STBD1. Does expression of STBD1 W203C mutant in STBD1 KO cells also accelerate tumor growth?

Ans: We highly appreciate these insightful suggestions! We performed multiple experiments as suggested. Multiple new cell lines were generated, including the shSTBD1 HCT116 cells were rescued with shRNA-resistant STBD1 WT (shSTBD1/WT), W203C (shSTBD1/W203C), or empty vector (shSTBD1/plvx neo). We found:

- (1) shSTBD1/WT, but not shSTBD1/W203C, significantly decreases cell growth (Supplementary Fig. 6a-b).
- (2) shSTBD1/WT pronouncedly suppresses tumor growth in the tumor xenograft model, in comparison with control (plvx neo) and shSTBD1/W203C (Fig. 4o-p).
- (3) Analysis of RNA-Seq results reveals that STBD1 W203C and shSTBD1 affect many similar pathways. Eight out of 14 pathways, including glycolysis/gluconeogenesis, enriched in the shSTBD1 vs. shControl cells are also found in the ones that are differentially affected in shSTBD1/W203C vs. shSTBD1/WT cells (Supplementary Fig. 8a-d).
- (4) Both STBD1 W203C and depletion of STBD1 affect the expression of genes in the glycolysis pathway (Fig. 5e).
- (5) Both STBD1 W203C and depletion of STBD1 promote the expression of c-Myc (Supplementary Fig. 9).

Collectively, our results indicate that STBD1 has potential tumor suppressive activity through interacting with LC3B and participating in glycolysis. STBD1 W203C mutation leads to the loss of normal functions of STBD1, similar to shSTBD1.

8. The connection between STBD1 and Myc is difficult to reconcile. It is unclear as to why myc levels change in response to STBD1 knockdown or expression of the W203C mutant. Can the authors provide an explanation as to how and why this occurs?

Ans: c-Myc is a well-characterized marker of multiple cancer hallmarks including evasion of anti-growth signaling, sustained proliferation and cell death. Since STBD1 has potential tumor-suppressive activities, we are interested in determining which cancer hallmark traits are affected by STBD1. From a well-curated database named HOC, we obtained 377 known cancer hallmark genes, and mapped them to our transcriptomic data with or without STBD1 depletion. The enrichment analyses

demonstrated that 5 cancer hallmarks might be regulated by STBD1, including sustained proliferation, genome instability, cell death, invasion and metastasis, and metabolism. As c-Myc associates with two cancer hallmarks regulated by STBD1, sustained proliferation and cell death, we suspected that STBD1 may affect the expression of c-Myc. Indeed, depletion of STBD1 dramatically increases the expression of c-Myc (new Fig. 5f, 5g and Supplementary Fig. 9).

To further validate our approach, we tested two additional hallmark genes, NFKB1 and AKT1, which are predicted to be affected by STBD1. We found that STBD1 depletion leads to up-regulation of oncogene AKT1, whereas suppresses the expression of tumor suppressor NFKB1 (Fig. 5g). Thus, STBD1 suppresses tumor growth through inhibiting multiple cancer hallmark traits.

9. Related to the previous point, a direct role for STBD1 in regulation of metabolism or expression of metabolic genes is not convincingly established – the data as presented is largely correlative. For instance, is it possible that the increased glycolysis associated with STBD1 loss or expression of the W203C mutant may be indirect – ie associated with changes in Myc levels? If so additional mechanistic insight into how STBD1 changes Myc activity is important to establish.

Ans: We have tried our best, but could not accurately predict any direct relations between c-Myc and STBD1, using available bioinformatic tools. In current stage, our algorithms including pLIRm and pLAM in iCAL pipeline can only prioritize potentially cancer-associated LIR-containing proteins (LIRCPs) that carry single point mutations within the LIR motif. Accurate prediction and demonstration of the connection between STBD1 and c-Myc, although potentially important, is out of the scope of our current study. However, our results presented above strongly supported that STBD1 depletion or mutation promote the acquisition of multiple cancer hallmark traits.

10. Overall the study delves more so into the functional significance of STBD1 rather than a specific role for the LIR mutant and its effect on tumorigenesis, metabolism and autophagy.

Ans: Thanks a lot for the suggestions. We have taken your comments very seriously, and performed multiple sets of experiments to demonstrate the importance of the LIR mutation in STBD1. As illustrated in Fig. 4o-p, Fig. 5e-g, Supplementary Fig. 6a-b, 8d, 9, we now provide multiple lines of new evidence that the LIR mutation in STBD1 affects tumor growth, expression of genes critical for glycolysis, and autophagy.

Reviewer #2:

1. The manuscript of Han et al describes a novel bioinformatics approach to identify autophagy targeted proteins, focuses on those that are mutated in cancer, and experimentally validates and assess one of the key predictions, STBD1. The manuscript is well written compared to the complexity of the study, though there are a few minor issues with the text listed below. The topic is very timely, and the applied approach extends our current knowledge. The assessment of the data has been carried out properly, except for a few points in relation for example to how the cancer-associated mutations were defined and used (see details below).

Major points:

- Title does not represent fully the study. It should be more specific to the presented work.

Ans: Based on your suggestion, we changed the title into “Model-based analysis uncovers mutations that change autophagy selectivity in cancer cells”.

2. The abstract does not contain the key results properly in relation to the STBD1 experiments. Ie, the paper contains much more than what is written currently in the abstract.

Ans: Thanks a lot for the suggestions! We have modified the abstract as suggested.

3. Throughout the manuscript abbreviation usage is not ideal. There are much more abbreviations than needed, any many are not standard ones. For example “AA” as autophagy-associated (AA) proteins is not really common, and confusing with ATG proteins, even if AA meant to cover more. I suggest reducing the number of abbreviated terms.

Ans: The abbreviation “AA” was replaced by ATG genes/proteins and autophagy regulators throughout the manuscript. Other non-commonly or less used abbreviations were removed.

4. The mutation data, while described in detail in the Methods section, is not always clear in the Results section which one was used and why. For example in the Results section: “Next, we mapped known cancer mutations to potential human LIRCPs “. In the Methods there are three mutation sources, which one was used here, and why?

Ans: In this study, all missense single nucleotide variants (SNVs) collected from TCGA, ICGC and COSMIC databases were used. The three data sets were highly redundant, and we merged them into a single data set and cleared the redundancy before use. We revised the manuscript to clarify this point as below:

Page 9, paragraph 2, added,

“From The Cancer Genome Atlas (TCGA) ²², International Cancer Genome Consortium (ICGC) ²³ and Catalogue of Somatic Mutations in Cancer (COSMIC) ²⁴, we obtain 2,963,952 non-redundant missense single nucleotide variants (SNVs).”

Page 32, paragraph 1, changed,

“...In total, we obtain 1,898,302, 4,306,716 and 5,806,067 human missense SNVs from TCGA, ICGC and COSMIC databases, respectively. We merge the three data sets together, and there are 2,963,952 unique missense SNVs reserved after redundancy clearance.”

5. Also, a more important point currently not covered is what was the mutational rate for the identified LAMs? How frequent they are in the different cancer types?

Ans: We agree with your opinion, and added the information of cancer type(s), numbers of affected cases against all cases, and percentile for each identified LAM if available (Supplementary Data 3). For COSMIC mutations, functional impacts predicted by FATHMM, a Hidden Markov Model (HMM)-based web-server to predict the functional impacts of coding or non-coding variants, were also present.

6. COSMIC contains for example so many passenger mutations that should not be considered in general as cancer-associated. Therefore, more data presentation and quality control are needed here, before applying their new model.

Ans: In the new Supplementary Data 3, there were 94 identified LAMs derived from the COSMIC database. From the functional impacts of these mutations predicted by FATHMM, it could be found that 87 (92.6%) COSMIC-derived LAMs were annotated as “Pathogenic” (FATHMM score > 0.8). Thus, this result indicated that cancer driver mutations might be truly enriched in identified LAMs.

7. At some parts, the data integration is not always justified or clear. For example, “The pan-cancer analysis revealed that SNVs, RNA expressions, and DNA methylation levels of 19, 101 and 109 LIRCPs are statistically associated with human cancer (Fig. 3a).” Why these data is relevant when LIR motif based associations would affect the protein level and not the regulation of the respected genes? If external data is aimed to use as in silico confirmation than proteomics difference for example from the Human Protein Atlas’s cancer dataset would be much more adequate.

Ans: To ensure the data quality, all computational analyses in this study were re-performed, and in total we identified 222 potential LIR motif-associated mutations (LAMs) in 148 proteins (Supplementary Data 3). We apologize for this inconvenience. Indeed, *bona fide* LAMs will change LIR motifs and affect the protein expressions, which should significantly correlate with clinical outcomes under the survival analysis to indicate potential roles in tumorigenesis. However, we cannot get a high-quality data

set of protein expressions in cancer. For example, in The Human Protein Atlas (HPA, <https://www.proteinatlas.org/>), exact protein expression values were not provided in the file “pathology.tsv.zip” (<https://www.proteinatlas.org/about/download>). Instead, expression profiles for proteins in human tumor tissues based on immunohistochemistry were annotated for different staining levels ("High", "Medium", "Low" & "Not detected"). The records of STBD1 were shown as below:

Gene Name	UniProt ID	Cancer type	Hig h	Mediu m	Lo w	Not detected
STBD1	O95210	breast cancer	0	1	2	9
STBD1	O95210	carcinoid	2	0	1	1
STBD1	O95210	cervical cancer	0	2	1	8
STBD1	O95210	colorectal cancer	0	2	3	5
STBD1	O95210	endometrial cancer	0	0	0	11
STBD1	O95210	glioma	0	1	3	6
STBD1	O95210	head and neck cancer	0	1	1	1
STBD1	O95210	liver cancer	4	2	4	2
STBD1	O95210	lung cancer	0	4	5	3
STBD1	O95210	lymphoma	0	2	7	3
STBD1	O95210	melanoma	1	6	0	4
STBD1	O95210	ovarian cancer	0	2	1	8
STBD1	O95210	pancreatic cancer	0	4	3	2
STBD1	O95210	prostate cancer	0	0	6	5
STBD1	O95210	renal cancer	0	5	2	5
STBD1	O95210	skin cancer	0	1	4	6
STBD1	O95210	stomach cancer	0	1	1	9
STBD1	O95210	testis cancer	0	0	1	11
STBD1	O95210	thyroid cancer	0	0	2	1
STBD1	O95210	urothelial cancer	0	3	1	7

Due to the limited information of protein expressions in cancer, we have to use SNVs, RNA-seq and DNA methylation data to probe the potential correlations between LIRCPs and cancer (Supplementary Data 4). It can be expected that such an information will be not as efficient as directing using the proteomic data, however, it still provided very useful candidates for our further experiments, and helped us to validate the role of STBD1 in tumorigenesis.

8. In general, the whole “LAM-containing LIRCPs play a potential role in human cancer “ section is out of scope or focus of this study. While relevance in cancer is an important thing to validate, these omics layer provide this just indirectly, not in relation to the LAMs.

Ans: As a pure bioinformatician, one of the corresponding authors, Yu Xue, definitely

agree with your opinion. However, his collaborative partners working in cancer biology insisted that such a systematic modeling was helpful for biologists to understand the potential roles of LAMs and LIRCPs in human cancer, and highlighted the importance of the study. We also consulted Dr. Garcia-Fernandez, the editor for this manuscript, who suggests that we should keep this section.

9. *The section on EHMT2, ERCC6, and BRAF are inserted in the middle of the STBD1 analysis and validation. Probably it would be better before the STBD1.*

Ans: Thanks for the excellent suggestion. We have modified this section as suggested.

10. *Also, from the current brief text it is not clear what the role of the newly discovered LIR motifs is in EHMT2, ERCC6, and BRAF? Especially the BRAF is very relevant to cancer biology. This should be presented better, or removed for the sake of the focus of the study.*

Ans: We have modified this section by putting the data on EHMT2, ERCC6, and BRAF before STBD1. For the sake of time, we spent most of our effort in studying STBD1. In the revised manuscript, we emphasize several times that BRAF and many other genes discovered in our study are worthy of further investigation.

11. *There is a network reconstruction work in the study, which is needed in principle but was not inserted, justified and carried out properly. This LIRCP-regulating network contains protein-protein interactions (PPIs) and transcriptional regulations among the 151 AA proteins. Why these interactions were added? This reconstruction study should be in the Results but currently it is referred only in the Discussion and the Methods. And additional issue is that the layout and visualization of the network on Figure 7 are not informative, unclear and not following best practices in the field (for example, please check the guidance here <https://doi.org/10.1371/journal.pcbi.1007244>)*

Ans: Thank you very much for your suggestion. We have put this section into Results entitled “A LIRCP-regulating network links autophagy selectivity and tumorigenesis”. Both PPIs and transcriptional regulations were considered, because both two mechanisms are important in regulating autophagy. We revised the manuscript as below:

Page 22, paragraph 2, added,

“A LIRCP-regulating network links autophagy selectivity and tumorigenesis

Understanding the mechanisms whereby the autophagy network interfaces with cancer is a long-standing challenge. The central questions include whether the autophagy pathways are targets for recurring molecular alteration in human cancer, and which pathways are targeted^{1, 2}. To identify the autophagy pathways perturbed in human cancer, we model a LIRCP-regulating network by integrating protein-protein interactions (PPIs) and transcriptional regulations among the 148 identified LIRCPs, 7 LC3 proteins and 14 proteins regulated by STBD1, since both mechanisms are

important for regulating autophagy^{32, 33, 34, 35} (Fig. 7).

Based on the functional annotations in UniProt³⁶, we classify 148 LIRCPs into 9 classes, including apoptosis-associated events, autophagic vacuole assembly, cell cycle/proliferation, small GTPase-associated signaling, inflammatory/immune response, metabolic pathways, PI3K/AKT/mTOR signaling, biomolecule/vesicle transport, and glycolysis. The 7 LC3 proteins are categorized into the class of autophagic vacuole assembly, based on their important role in autophagosome formation. The 14 downstream proteins of STBD1 are also included. Known or predicted PPIs and transcriptional regulations between transcription factors and target genes are integrated from 8 public databases, including ARN³², BioGrid³⁷, IID³⁸, inBio MapTM³⁹, Mentha⁴⁰, HINT⁴¹, iRefIndex⁴² and PINA⁴³. In total, we obtain 2,204 PPIs and 91 transcriptional regulations for the 169 proteins, in which known cancer hallmark proteins are also indicated³⁵ (Fig. 7). From the network, it can be found how LIRCPs affect human cancer through the 9 functional aspects, and highlight the functional importance of STBD1 in inhibiting cancer growth through modulating glycopagy (Fig. 7). Our work indicates cancer cells frequently alter autophagy selectivity for survival.”

Page 47, paragraph 2, changed,

“From the 8 public databases including ARN³², BioGrid³⁷, IID³⁸, inBio MapTM³⁹, Mentha⁴⁰, HINT⁴¹, iRefIndex⁴² and PINA⁴³, we collect 1,771,193 PPIs and 131,541 transcriptional regulations of 18,839 human proteins from these databases, and map them to the 148 identified LIRCPs, 7 LC3 proteins and 14 proteins regulated by STBD1. Among the 91 transcriptional regulations, there are 88 for STAT1 and 3 for STAT3, respectively. Both two transcription factors, STAT1 and STAT3, are important autophagy regulators^{73, 74}. Similar to previous studies^{33, 34}, the LIRCP-regulating network is constructed and visualized with Cytoscape 3.7.2 software package⁷⁵.”

12. The discussion on the LIR prediction model is missing, more details on the benchmarking with other existing methods is needed, not only that the coverage of the new method is higher.

Ans: Based on your concern, we perform additional comparisons of pLIRm and other existing methods, and the results are present in a new Supplementary Note 1 entitled “Additional comparisons of pLIRm and other existing methods”.

13. How many LAMs can be found with the existing methods?

Ans: iLIR, hfAIM and other motif-based methods are developed for predicting potential LIR motifs. To the best of our knowledge, pLAM is the only algorithm to predict potential LAMs.

14. How many putative LIR motifs (from iLIR database and iLIR@viral, which were

correctly left out in the original data curation) can be found/confirmed with the new extended algorithm.

Ans: In iLIR database and iLIR@viral database, all canonical LIRs (cLIRs) were annotated, although iLIR prediction scores were also presented. Using a same threshold of $Sp = 90\%$, we compare the results of iLIR and pLIRm, and find that only a moderate proportion of iLIR-derived hits are covered by pLIRm. The poor overlap between iLIR and pLIRm might be attributed to a small training data set used in iLIR. The corresponding results are described in the Supplementary Note 1.

15. *The “A Modified GPS Algorithm “section is difficult to understand. More explanation is needed to understand and asses this part.*

Ans: Based on your concern, we add more details on the GPS algorithm by revising the manuscript as below:

Page 28, paragraph 3, changed,

“In 2004, we developed the GPS 1.0 algorithm for prediction of kinase kinase-specific phosphorylation sites ⁵⁷. Based on a hypothesis that similar short peptides might share similar biological properties and functions, we used an amino acid substitution matrix, e.g., BLOSUM62, to measure the sequence similarity among short peptides around known or putative phosphorylation sites ⁵⁷. We adopted this basic scoring strategy in all versions of GPS algorithms, and incorporated more methods to improve the accuracy in later versions ⁵⁸. In GPS 5.0 (<http://gps.biocuckoo.cn/>) ¹⁷, we developed two additional approaches including PWD and SMO in order to improve the accuracy, besides the basic scoring strategy. PWD could efficiently optimize the position-specific weight values of short peptides around phosphorylation sites, whereas SMO could rapidly determine the scoring matrix. We used the PLR algorithm with the ridge (L2) penalty to optimize parameters ¹⁷.

Here, we **modify** the original GPS 5.0 algorithm to comprise two parts. In the part of the basic scoring strategy, **we measure the average similarity score (S) of a given LMP(7, 7) item** against all known LMP(7, 7) entries in positive data as below:”

16. *A potential key issue is that according to the Methods from the human proteome set, the authors detected 19,577 proteins containing at least one tetapeptide that follows the cLIR motif. As they checked 20,659 proteins, it would mean that the authors have found these motifs in nearly all human proteins which is worrying. Even if they apply a model to filter false positives, this initial step and data could question their approach and the results, thus this should be addressed.*

Ans: Here, we confirm that we truly find 19,577 proteins containing at least one tetapeptide cLIRs that follow the sequence pattern [FWY]XX[LIV]. This motif is very short and loosely defined. Thus, it's not surprise that over 94.8% of all human proteins

contain this tetapeptide. We mapped non-redundant missense SNVs in cancer to this data, and identify 842,789 potential LAMs located in or around 238,840 cLIRs of 18,806 human proteins. It can be expected that most of these initial LAMs will be non-functional, and this data is a high-quality data set for us to estimate the global distribution of potential impacts of missense SNVs. From the results, it can be found that most of the initial LAMs have very slight influence before and after mutation. Using pLAM, we only prioritize 222 potential LAMs in 148 LIRCPs as useful candidates for further analysis. If the size of the initial data is small, the estimation of the global distribution might be biased and not accurate enough.

17. *Technical details:*

- On page 25, add specific Uniprot release number and or date of download – year is not enough.

Ans: Sure, and we change the corresponding description as “...Then, we map known LIR motifs to primary protein sequences downloaded from UniProt database ³⁶ to pinpoint their exact positions (On October 17, 2019).”

18. *On page 25, It is not clear what was the initial rational to define the short flanking peptides and the LMP(7, 7).*

Ans: We adopted a LMP(7, 7) with a length of 18 aa to balance the training time and accuracy. Longer peptides will cost more training time. We change the corresponding description as “...with a total length of 18 aa to balance the training time and accuracy.”

19. *On page 25, It is also not clear how the positive and the negative datasets were compiled from this sentence: “we regarded LMP(7, 7) entries derived from all known cLIR and aLIR motifs as positive data, and we took LMP(7, 7) items around other putative cLIR motifs in the same proteins as negative data.”*

Ans: We add a new sheet “Training data set” in Supplementary Data 1 to present both positive and negative data sets, which can clarify the ambiguous point on the preparation of the benchmark data set.

20. *Where is the result of the “Performance Measurements” mentioned in the Methods (page 26)?*

Ans: We apologize for this convenience, and add a new Supplementary Data 2 to present the performance measurements of pLIRm under high, medium and low thresholds.

21. *On page 26, “Recently, we improved our previously-developed GPS algorithm from 2.1 to 5.0...” – it is not clear what 2.1 to 5.0 means.*

Ans: This unclear description is deleted, and the corresponding paragraph is re-written to present more details on the GPS algorithm.

22. *In the end of the same sentence on page 26 "in order to predict kinase-specific phosphorylation sites (<http://gps.biocuckoo.cn/>, published elsewhere)." – please add the exact reference.*

Ans: The reference has been added.

23. *Minor:*

- *In the abstract: "Further analysis confirms that LAMs in ATG4B, EHMT2 and BRAF that can alter interactions with LC3 and/or autophagic activities." – remove "that"*

Ans: The abstract has been re-written, and this error is cleared.

24. *In the abstract: "Unexpectedly, STBD1, a poorly-characterized protein, inhibits tumor growth via modulating glycogen autophagy, while its cancer-linked mutation abolishes the cancer inhibitory function." I suggest rewriting as the "unexpectedly" refers to the second part of the sentence.*

Ans: Thank for the suggestion. We have re-written the abstract.

25. *In the abstract: "... provides a fundamental framework to uncover the molecular landscape that drives carcinogenesis via modulating autophagy selectivity." – I do not think it is a "molecular landscape", rather a "molecular mechanism".*

Ans: The Abstract has been re-written, and the nomenclature "molecular landscape" is not used any longer.

26. *In the Introduction: "LIRCPs " is not introduced or described what it means.*

Ans: In the introduction, we changed the corresponding description as "...Using iCAL, we have identified 148 LIR-containing proteins (LIRCPs) that carry single point mutations within the LIR motif, including some well-established ATG genes and autophagy regulators as well as many novel candidate genes." in Page 6, paragraph 1.

27. *"Our benchmark data set was much larger than iLIR (Kalvari et al., 2014) and hfAIM (Xie et al., 2016), which only collected 27 and 36 known LIR motifs, respectively " – Probably add details or reference to Fig2a where the authors' benchmark is quantified properly.*

Ans: The legend of Fig. 2a has been changed as "a) A comparison of known LIR motifs and corresponding proteins collected by iLIR¹⁹, hfAIM²⁰ and pLIRm, as well as the

distribution of our collected data in *H. sapiens* and *S. cerevisiae* and other species (Supplementary Data 1).”

28. “we identified 233 potential LAMs that significantly change 177 cLIR motifs in 151 LIRCPs, including 64 Type I and 169 Type II LAMs” – Please rewrite as “including” is not in a proper place of the sentence.

Ans: We changed the corresponding description as “...In total, we identify 222 potential LAMs including 60 Type I and 162 Type II LAMs that significantly change 172 cLIR motifs in 148 LIRCPs (Fig. 2d and Supplementary Data 3).” in Page 9, paragraph 2.

29. “indicating that a strong correlation between autophagy and human cancer “ – remove “that”

Ans: We have fixed this.

30. On page 17, “differentially expression genes (DEGs)” – change to “differentially expressed genes”

Ans: We have fixed this.

31. On page 22, “Glycophagyis” -> Glycophagy is”

Ans: We have fixed this.

32. On page 27., “tothe” -> “to the”

Ans: We have fixed this.

33. On Figure 1: There is a typo in “tumor proliferation”

Ans: We have fixed the typo.

REVIEWER COMMENTS

Reviewer #1 (Remarks to the Author):

The authors have made a concerted effort to address all my comments. In particular the inclusion of significant studies comparing the WT and LIR mutant of STBD1 in in vitro and in vivo experiments is a strength. I only have minor suggestions relating to narrative.

I believe the study would read better if the authors emphasize the pro-tumorigenic activity of suppressing STBD1 – either via knockdown or expression of the mutant. Similarly, it would be useful to discuss why glycophyagy would be detrimental or less beneficial to some cancer cells, as implied by the authors finding that STBD1 is suppressed.

Finally, in the text which states that the authors test whether “STBD1 depletion influences genome instability and invasion and metastasis” by looking at AKT1 and NFkB1 is not justified. Changes in AKT1 or NFkB1 do not confer information regarding genome instability, invasion or metastasis in the absence of specific assays to test these features. The sentence should be changed to indicate that other pro-tumorigenic pathways (AKT1 levels) are upregulated. If possible it would be useful to actually probe AKT1 phosphorylation or its downstream targets to more directly indicate that the pathway is more active in STBD1 knockdown cells. However, again this is largely correlative.

The abstract needs to be corrected for spelling and grammar.

Reviewer #2 (Remarks to the Author):

I went through all the responses given to my previous comments in the response letter, and checked it in the revised manuscript. The authors elegantly and correctly addressed all my previous concerns. I was really pleased to see the new Figure 7, and all the corrections they have added during the revision. I do not have any further comments or concerns for this work to be published. I thank the authors for the clear and high-quality revision.

Tamas Korcsmaros

Reviewer #3 (Remarks to the Author):

Overview

The authors applied metabolomics to find that STBD1 knockdown cells have increased glucose utilization as indicated by: 1) increased abundances and ¹³C-labeling of glycolytic intermediates, PPP intermediates, and nucleotides; and 2) decreased in media glucose. This is a straightforward application of metabolomics with clear results that complement the rest of the paper. Based on expert review, addressing the

comments below will improve the results and discussion of Figure 6, Supplementary Figure 10, and Supplementary Data 8.

Major comments

- Fig 6b: Is the heatmap displaying ratio of averages? It would be better to display individual samples; otherwise barplots with error bars should be used instead to give some indication of the variance in the data. Additionally, this heatmap appears to combine data from both metabolite profiling and ¹³C-labeling experiments; this makes it rather confusing to interpret. The authors should consider separating data for the metabolite profiling and ¹³C-labeling experiments into two figures. Also, for ¹³C labeling data, the specific labeled isotopologs (M1, M2 etc) should be indicated.
- Fig 6d: Specific labeled isotopologues should be indicated.
- Fig 6g and 6h: Strictly speaking, MTT assays measure reducing power and the results may be influenced by differences in metabolic activity/NADH production in the cells. This is especially relevant here since the authors found STBD1 depletion to influence metabolism. This caveat should be mentioned in the discussion.
- The authors additionally found that STBD1 knockdown cells have decreased TCA cycle metabolites. This is interesting and should be discussed further. If STBD1 knockdown cells have enhanced glycolysis and consume more glucose, but do not produce more lactate, one would expect glycolytic intermediates to be feeding into the TCA cycle but this does not seem to be the case. Where are the products of increased glycolysis going? Is the glucose all being shunted into nucleotide biosynthesis?
- In the Methods section, the following information should be included:
 1. How many cells were seeded, and how long were they cultured (e.g. 24 h) before extraction?
 2. How was metabolite data normalized?
 3. What do 'relative concentrations' mean: was absolute quantification performed for either intracellular metabolites or media glucose and lactate levels, then normalized to control samples? Or do relative concentrations simply mean relative peak areas?

Minor comment

- Page 21, lines 8-10: "Subsequently, more ¹³C6-glucose is incorporated into the purine (AMP and GDP) and pyrimidine (UMP, CDP and CTP) ring, required to make RNA and DNA in proliferating cells in shSTBD1 cells (Fig. 6b)." It is confusing to bring in ¹³C labeling results here, since both preceding and following paragraphs are concerning metabolite levels. This should be moved to later in the paragraph, e.g. after "These results are further supported by metabolic flux analysis using stable ¹³C6-glucose labeling."

Detailed Responses to Reviewers' Comments

Reviewer #1:

1. The authors have made a concerted effort to address all my comments. In particular the inclusion of significant studies comparing the WT and LIR mutant of STBD1 in in vitro and in vivo experiments is a strength. I only have minor suggestions relating to narrative.

I believe the study would read better if the authors emphasize the pro-tumorigenic activity of suppressing STBD1 – either via knockdown or expression of the mutant. Similarly, it would be useful to discuss why glycolysis would be detrimental or less beneficial to some cancer cells, as implied by the authors finding that STBD1 is suppressed.

Ans: Suggestions are well-taken. We have added one paragraph in the discussion section to discuss why suppressing STBD1 and glycolysis contribute to tumorigenesis.

Page 25, paragraph 2, added,

“Why does the inhibition of glycolysis contribute to tumorigenesis? Glycogen is degraded via two major pathways: the cytosolic pathway that decomposes glycogen into glucose-1-phosphate and glucose, and glycolysis that decomposes glycogen into glucose. Therefore, it is expected that glycolysis inhibition could cause metabolic reprogramming. Indeed, we find that the depletion of STBD1 increases expression of multiple key glycolytic enzymes, and enhances the TCA cycle and nucleotide biosynthesis. Suppression of STBD1 – either via knockdown or expression of the mutant – also promote the expression of multiple cancer hallmark genes, including c-Myc, NFKB1 and AKT1, although the exact underlying mechanisms remain to be determined...”

2. Finally, in the text which states that the authors test whether “STBD1 depletion influences genome instability and invasion and metastasis” by looking at AKT1 and NFKB1 is not justified. Changes in AKT1 or NFKB1 do not confer information regarding genome instability, invasion or metastasis in the absence of specific assays to test these features. The sentence should be changed to indicate that other pro-tumorigenic pathways (AKT1 levels) are upregulated.

Ans: Thanks for the suggestion. We have modified the corresponding sentence in Page 20, paragraph 1, as “...indicating that other pro-tumorigenic pathways are also upregulated.”

3. *If possible it would be useful to actually probe AKT1 phosphorylation or its downstream targets to more directly indicate that the pathway is more active in STBD1 knockdown cells. However, again this is largely correlative.*

Ans: We have measured AKT1 phosphorylation as suggested. In two of our three experiments, we found that STBD1 knockdown led to an increase of phosphorylation of AKT1 at Ser473 (left and central panel); however, in our third experiment, we did not observe a significant increase (right panel). As a result, we do not include this piece of data in our manuscript. However, we have consistently observed that STBD1 depletion leads to up-regulation of oncogene AKT1.

4. *The abstract needs to be corrected for spelling and grammar.*

Ans: The suggestion is well-taken. We have fixed the grammar.

Reviewer #3:

1. Overview

The authors applied metabolomics to find that STBD1 knockdown cells have increased glucose utilization as indicated by: 1) increased abundances and ¹³C-labeling of glycolytic intermediates, PPP intermediates, and nucleotides; and 2) decreased in media glucose. This is a straightforward application of metabolomics with clear results that complement the rest of the paper. Based on expert review, addressing the comments below will improve the results and discussion of Figure 6, Supplementary Figure 10,

and Supplementary Data 8.

Ans: Thank you very much for your excellent suggestions. Based on your comments, we have re-analyzed the data of targeted metabolomic profiling and isotope tracing analysis, and changed the original Fig. 6 and Supplementary Data 8, as well as the legend of Fig. 6. The first paragraph of the section “STBD1 depletion promotes glycolysis in cancer cells” in Page 20 has been carefully re-phrased.

2. Major comments

• *Fig 6b: Is the heatmap displaying ratio of averages? It would be better to display individual samples; otherwise barplots with error bars should be used instead to give some indication of the variance in the data. Additionally, this heatmap appears to combine data from both metabolite profiling and 13C-labeling experiments; this makes it rather confusing to interpret. The authors should consider separating data for the metabolite profiling and 13C-labeling experiments into two figures. Also, for 13C labeling data, the specific labeled isotopologues (M1, M2 etc) should be indicated.*

Ans: In the revision, data from metabolic profiling and isotope labeling analysis is separated. The metabolites detected by the targeted metabolomic profiling are shown in Fig. 6b for several major pathways including glycolysis, tricarboxylic acid (TCA) cycle, purine metabolism, and pyrimidine metabolism. In the heatmap, the levels of metabolites are shown for individual samples, after a z -score normalization. For $^{13}\text{C}_6$ -glucose labeling data, barplots are shown in Fig. 6c-f for key metabolites in the above 4 pathways, and the specific labeled isotopologues which represent specific pathways have been indicated. We added corresponding descriptions on the z -score method by revising the manuscript as below:

Page 46, paragraph 1, added,

“...For plotting the heatmap, the original expression levels of each metabolite in the 6 samples are normalized using the z -score transformation, one of the mostly used normalization methods⁷³. The mean expression value μ and standard deviation (SD) δ are calculated. For the metabolite i with the expression level of x_i , its normalized z -score is calculated as below:

$$z_i = \frac{x_i - \mu}{\delta}$$

”

3. *Fig 6d: Specific labeled isotopologues should be indicated.*

Ans: Specific labeled isotopologues have been added in Fig. 6c-g.

4. Fig 6g and 6h: Strictly speaking, MTT assays measure reducing power and the results may be influenced by differences in metabolic activity/NADH production in the cells. This is especially relevant here since the authors found STBD1 depletion to influence metabolism. This caveat should be mentioned in the discussion.

Ans: Suggestions are well-taken. We have included a paragraph in the discussion section to emphasize the caveat of MTT assay when metabolic states of cells are altered. We used both the MTT assay and Ki67 to measure cell proliferation in our experiments. This information will be valuable for the whole community.

Page 26, paragraph 1, added,

“...We show that STBD1 suppression promotes cancer cell proliferation, detected by the MTT assay and the proliferation marker Ki67. It should be noted that the MTT assay measures reducing power, in particular, NADH in mammalian cells. Therefore, cautions must be taken to use the MTT assay when metabolic states of cells are altered. Other methods of detecting cell proliferation, such as detection of Ki67 and/or BrdU labeling, should be used at the same time.”

5. The authors additionally found that STBD1 knockdown cells have decreased TCA cycle metabolites. This is interesting and should be discussed further. If STBD1 knockdown cells have enhanced glycolysis and consume more glucose, but do not produce more lactate, one would expect glycolytic intermediates to be feeding into the TCA cycle but this does not seem to be the case. Where are the products of increased glycolysis going? Is the glucose all being shunted into nucleotide biosynthesis?

Ans: Thanks for the suggestion of separating data for the metabolite profiling and ¹³C-labeling experiments. Now the conclusion about glucose metabolism is much more straightforward. Since isotope labeling data is more interpretable and precise to explain pathway regulation in cancer cells, in the revision, we mainly referred data from ¹³C-glucose labeling experiments. We did conclude that in STBD1 knockdown cells glycolysis intermediates fed into both TCA cycle and nucleotide biosynthesis, indicating oxidative phosphorylation and generation of RNA/DNA are more active in shSTBD1 cells. We revised the manuscript as below:

Page 20, paragraph 2, changed,

“The above findings suggest that STBD1 depletion **potentially** leads to metabolic reprogramming. To probe such changes, we **first** perform **a targeted** metabolomic

profiling of shControl and shSTBD1 HCT116 cells, each with three biological replicates (Fig. 6a, Supplementary Data 8). More than 200 metabolites in various metabolic pathways, including glycolysis, tricarboxylic acid (TCA) cycle, purine metabolism, pyrimidine metabolism, and amino acids, are observed (Fig. 6b, Supplementary Data 8). To trace the glycolytic flow in cancer cells, we further perform an isotope tracing analysis using stable $^{13}\text{C}_6$ -glucose labeling (Fig. 6c-f, Supplementary Data 8). Knockdown of STBD1 leads to increased glycolytic intermediates, as represented by 3-Phosphoglycerate/2-Phosphoglycerate (3-PG/2-PG) $m + 3$, phosphoenolpyruvate (PEP) $m + 3$, and pyruvate $m + 3$ in the glycolysis pathway (Fig. 6c). Meanwhile, enhanced glucose metabolism into TCA cycle, e.g. citrate $m + 2$, aconitate $m + 2$, isocitrate $m + 2$, and α -ketoglutarate (α -KG) (Fig. 6d), is observed, as well as nucleotide biosynthesis through pentose phosphate pathway, e.g., AMP $m + 5$ in purine metabolism and UMP $m + 5$ in pyrimidine metabolism (Fig. 6e-f). The unchanged intracellular level of lactate $m + 3$ (Fig. 6c) further confirms the enhancement of glucose metabolism is biased into oxidative phosphorylation and nucleotide biosynthesis, and the latter is required to make RNA and DNA in proliferating cells in shSTBD1 cells. The results are highly consistent with our observation in the transcriptomics (Fig. 6g). In contrast, most essential amino acids are not altered by knockdown of STBD1, based on the targeted metabolomic profiling (Supplementary Fig. 10a). Taken together, depletion of STBD1 leads to substantial reprogramming of glucose metabolism in cancer cells through enhanced glycolysis.”

6. In the Methods section, the following information should be included:

How many cells were seeded, and how long were they cultured (e.g. 24 h) before extraction?

Ans: Thanks for the suggestion, we have added these information.

Page 45, paragraph 2, added,

“HCT116 cells stably expressing shControl or shSTBD1 established as described above are cultured in DMEM supplemented with 10% FBS for 24 h. Then shControl and shSTBD1 HCT116 cells (1×10^7) are harvested and extracted using pre-chilled 80% (v/v) methanol.”

Page 46, paragraph 1, added,

“For $^{13}\text{C}_6$ -labeled metabolites analysis, shControl and shSTBD1 HCT116 cells are cultured in glucose-free DMEM (Gibco) supplemented with 2.25 g/L $^{13}\text{C}_6$ -glucose and 2.25 g/L unlabeled-glucose for 12 h. The cells (1×10^7) are then harvested, and extracted using pre-chilled 80% (v/v) methanol as above.”

7. *How was metabolite data normalized?*

Ans: Samples were normalized using cell numbers when re-dissolved using 80% MeOH prior to LC-MS/MS analysis. The injection volume of all sample was 3 μ L. In this case, all metabolites were measured under the same cell numbers for comparison. And we have added this information in the methods.

8. *What do 'relative concentrations' mean: was absolute quantification performed for either intracellular metabolites or media glucose and lactate levels, then normalized to control samples? Or do relative concentrations simply mean relative peak areas?*

Ans: Thanks a lot! The lactate and glucose concentration was first normalized to the cell protein concentration, and the relative concentration was then normalized to the shControl HCT116 cells. And we have added this information in the methods and figure legends.

9. *Minor comment*

• *Page 21, lines 8-10: "Subsequently, more $^{13}\text{C}_6$ -glucose is incorporated into the purine (AMP and GDP) and pyrimidine (UMP, CDP and CTP) ring, required to make RNA and DNA in proliferating cells in shSTBD1 cells (Fig. 6b)." It is confusing to bring in ^{13}C labeling results here, since both preceding and following paragraphs are concerning metabolite levels. This should be moved to later in the paragraph, e.g. after "These results are further supported by metabolic flux analysis using stable $^{13}\text{C}_6$ -glucose labeling."*

Ans: The first paragraph of the section "STBD1 depletion promotes glycolysis in cancer cells" in Page 20 has been considerably revised.